# A JAK/STAT-Pdk1-S6K axis bypasses systemic growth restrictions to promote regeneration

Ananthakrishnan Vijayakumar Maya [1,2,3], Lena Neuhaus[4,5,6], Liyne Nogay[1,2,3], Aakriti Singh [1,2,3], Lara Heckmann [1,2,3], Isabelle Grass[1,2], Jörg Büscher [7], Katrin Kierdorf [5,6,8] & Anne-Kathrin Classen [1,2,8] ✉

Inflammation triggers systemic growth restrictions, a process well characterised in tumour cachexia. Whether inflammatory tissue damage also induces growth restrictions, and how regenerating tissue overcome them, is less explored. Using a tissue damage model in *Drosophila*, we identify metabolic and signaling adaptations that both induce and bypass systemic growth restrictions. Expression of *eiger*, the *Drosophila* TNF-α homolog, in imaginal discs causes systemic insulin restriction and insulin resistance, reducing protein translation and proliferation in peripheral tissues. Regenerating cells overcome this by upregulating Pdk1, which is necessary and sufficient to promote protein translation via an Insulin/Akt-independent mechanism. JAK/STAT acts upstream to elevate Pdk1, defining a JAK/STAT-Pdk1-S6K axis essential for regenerative proliferation. Regenerating cells also upregulate amino acid transporters and rely on mTORC1. Similar signatures in Ras$^{V12}$, scrib tumors indicate that tumors co-opt these pathways to sustain growth under insulin restriction. This physiological program thus integrates systemic nutrient mobilization and local metabolic reprogramming, with implications for tissue repair but also pathologies, such as chronic wounds and cancer.

Tissue damage and inflammation trigger a dynamic interplay between cellular signals and cell behaviours to promote repair and regeneration. Proper orchestration of these responses is crucial, as failure can lead to chronic wound healing pathologies or diseases like cancer[1–5]. To better understand these diseases, a wide range of recent studies aim to dissect the pathological reprogramming of relevant metabolic circuits[6–8]. However, a surprising gap exists in our knowledge about the precise metabolic adaptations employed by the normal physiological programmes of tissue repair and regeneration[1,9–11]. With this study, we aim to provide insight into the local and systemic metabolic adaptations during physiological tissue repair and regeneration.

Central to both metabolism and cellular growth are the Insulin/PI3K/Akt and mTORC1 signalling pathways, which are evolutionarily conserved from invertebrates to vertebrates. Both pathways converge on their shared effector ribosomal protein S6 kinase (S6K), which drives protein synthesis and cellular growth by activating protein translation. The Insulin/PI3K/Akt signalling branch is activated by binding of Insulin to its receptor, stimulating Phosphoinositide-3-kinase (PI3K) to produce Phosphatidylinositol (3,4,5)-trisphosphate

[1]Hilde-Mangold-Haus, University of Freiburg, Freiburg, Germany. [2]Faculty of Biology, University of Freiburg, Freiburg, Germany. [3]International Max Planck Research School for Epigenetics, Biophysics and Metabolism (IMPRS-EBM), Freiburg, Germany. [4]Spemann Graduate School of Biology and Medicine (SGBM), University of Freiburg, Freiburg, Germany. [5]Institute of Neuropathology, Faculty of Medicine, Medical Center, University of Freiburg, Freiburg, Germany. [6]Institute for Infection Prevention and Control, Faculty of Medicine, Medical Center, University of Freiburg, Freiburg, Germany. [7]Max Planck Institute of Immunobiology and Epigenetics, Freiburg, Germany. [8]Signaling Research Centres BIOSS and CIBSS, University of, Freiburg, Germany. ✉e-mail: anne.classen@biologie.uni-freiburg.de

(PIP3). PIP3 recruits Phosphoinositide-dependent kinase 1 (Pdk1) and activates Akt, which inhibits nuclear translocation of the transcription factor FOXO. Pdk1 phosphorylates S6K, initiating its activation, whereas optimal S6K activity requires an additional phosphorylation by mTORC1[12–16]. mTORC1 specifically responds to amino acid availability and is therefore central to anabolic growth[17]. While previous studies implicate a role for Insulin/PI3K/Akt and mTORC1 signalling in tissue repair processes, the precise metabolic adaptations remain to be investigated[18–21].

*Drosophila* models have advanced our understanding of tissue repair, regeneration and metabolism[22,23]. Specifically, studies in developing imaginal discs or the adult gut have highlighted the role of two key signalling pathways - JNK/AP1 and JAK/STAT - in repair and proliferation. These pathways coordinate a range of responses, from senescent-like cell cycle arrest in damaged cells to compensatory proliferation in adjacent cells[3,20,24–27]. Importantly, arrested, JNK-signalling cells produce Unpaired (Upd) cytokines, which activate JAK/STAT signalling in surrounding cells at the site of inflammatory damage, promoting survival and rapid regenerative proliferation[25,26,28–34]. The distinct functional demands of senescent and rapidly proliferating cells raise the question about how these distinct cell populations metabolically adapt to successfully support tissue repair.

Tissue repair and tumour development share striking similarities; in fact, tumours have long been compared to non-healing wounds[35]. Accordingly, *Drosophila* tumour models activate JNK/AP1 and JAK/STAT signalling, which promote tumour progression[30,36,37]. To support their growth, tumours secrete signalling molecules that initiate inter-organ signalling and systemic metabolic responses[38–40]. For instance, the Insulin-like peptide 8 (Dilp8), when secreted by tumours, disrupts hormone balance by reducing Ecdysone and Insulin production through direct effects on the ring gland and Insulin-producing cells (IPCs), with the effect of halting developmental progression of the tumour-bearing host[41,42]. Other cytokines, such as Ecdysone-inducible gene L2 (ImpL2), or the TNFα homologue Eiger (Egr), directly or indirectly trigger lipolysis and proteolysis to promote nutrient release from muscles as well as the fat body, an adipose tissue central for nutrient storage and energy homoeostasis[40,43–50]. Amino acids or sugars are released in this manner and subsequently absorbed by tumours, and facilitate their anabolic growth[51–53]. This inter-organ signalling network and metabolic state resembles cachexia, a clinical syndrome characterised by weight loss, muscle atrophy and fatigue, typically observed in chronic inflammatory conditions, including cancer[38–40,54,55]. While these studies reveal oncogenic metabolic signalling networks, the metabolic signalling networks employed during physiological tissue repair and regeneration remain poorly understood. Previous studies demonstrate that systemically acting cytokines may also be released upon tissue damage in the absence of oncogenic transformation[41,42,56–59], and fat body break-down changing Methionine, S-adenosylmethionine, and Kynurenine metabolism promotes imaginal disc regeneration[60,61]. However, the integration of local and systemic metabolic adaptations promoting physiological repair and regeneration remains poorly characterised. In our study, we combine genetic analysis, quantitative imaging, untargeted metabolomics and information from single-cell RNA sequencing data to outline the local and systemic adaptations that selectively support fast-proliferating cells during regeneration through an Insulin-independent JAK/STAT-Pdk1-S6K signalling axis.

## Results
### Regenerating cells maintain high levels of translation
To induce regeneration, we expressed the *Drosophila* homologue of TNF-α, known as Eiger (Egr), for a 24 h period within the imaginal wing pouch using the *rn-GAL4* driver[62]. As expected, *egr*-expression caused significant cell death, accompanied by the activation of the JNK/AP-1 activity reporter TRE-RFP (Fig. 1A–C and Supplementary Fig. S1A, B)[24,26,28,63]. The central disc region with high JNK signalling exhibited

markers of cellular senescence, including increased senescence-associated β-galactosidase activity (Supplementary Fig. S1C, D), upregulation of the matrix metalloprotease MMP-1 and cytokines of the Upd family, as well as a JNK-signalling induced cell cycle arrest in G2[24–26,32,64,65]. In contrast, cells within 40 μm surrounding this central JNK signalling domain were highly proliferative, as detected by EdU incorporation (Fig. 1A, B, D)[29,62]. To facilitate quantification of cell behaviours and account for disc size variation, we defined a conservative 20 μm band outside the high JNK signalling domain as the 'proliferative domain' (PD$^{egr}$) for the remainder of this study.

How do these proliferating cells meet their metabolic needs? To answer this question, we monitored protein translation using O-propargyl-puromycin (OPP)-incorporation assays[66]. This assay revealed uniform levels of protein synthesis throughout control wing imaginal discs. In *egr*-expressing discs, protein synthesis in the proliferative and the JNK signalling domains proceeded at levels similar to control discs (Fig. 1E–G). Interestingly, the notum of *egr*-expressing discs exhibited a significant decrease in OPP incorporation, which correlated with a marked reduction in EdU incorporation (Fig. 1G–M). The contrasting levels of protein synthesis, proliferation and signalling between different regions reveal that inflammatory damage induces at least three distinct cell populations with different cellular programmes: (1) a senescent cell population exhibiting high protein synthesis and JNK/AP-1 activity; (2) an cycling cell population exhibiting high protein synthesis and low JNK/AP-1 activation; and (3) notum cells exhibiting low protein synthesis, low JNK/AP-1 activity and slow cell cycling (Fig. 1N). While these observations mirror proliferation dynamics visualised in earlier studies[33,62,67], we wanted to better understand how these differences reflect a need to integrate local and systemic metabolic demands during regeneration.

### Tissue damage induces systemic insulin restriction
A reduction in protein translation rates in peripheral tissue domains like the notum suggested that insulin signalling, normally supporting protein translation through S6K activation, is reduced. Notably, cells with active JNK signalling express high levels of Dilp8, a known antagonist of insulin production by insulin-producing cells (IPCs) in the larval brain (Fig. 2A, B)[42,68]. Expression of *eiger* could thus limit anabolic growth despite nutrient intake by feeding[43,44,46,59,68–70]. To determine if *egr*-expressing larvae indeed restrict Insulin expression, we analysed the expression of the *Drosophila* Insulin-like peptides *dILP2* and *dILP5*[71]. We found that *dILP2* and *dILP5 expression* was significantly reduced in *egr*-expressing larvae, and approached levels seen in larvae starved for 24 h (Fig. 2C), a condition known to reduce insulin expression due nutrient limitation[71,72].

To understand if this reduction in *dILP2* and *dILP5* expression correlates with reduced Insulin signalling in peripheral tissues, we examined nuclear localisation of the dFOXO, a key downstream effector of canonical Insulin signalling[16,73]. We observed elevated nuclear dFOXO in the notum of *egr*-expressing discs (Fig. 2D–F), demonstrating that reduced *dILP2* and *dILP5* expression correlated with systemic attenuation of insulin signalling. This attenuation would explain the low rates of protein translation observed in the notum. In support of this conclusion, we found that low rates of protein translation in nota did not correlate with activation of JNK stress signalling or apoptosis, indicating that low Insulin signalling likely causes the observed reduction in protein synthesis (Supplementary Fig. S2A, B). Importantly, this systemic reduction in protein translation was also evident in other imaginal discs, such as the leg and the eye, and this effect was robustly reproduced by *eiger* expressed under the more restricted spatial pattern of *salm-GAL4* (Fig. 2G–T and Supplementary Fig. S2C–K). In all cases, the reduction in translational capacity correlated with decreased proliferation, with the developing eye showing reduced EdU incorporation overall and in the second mitotic wave, specifically (Supplementary Fig. S2H, I). Taken together, these findings

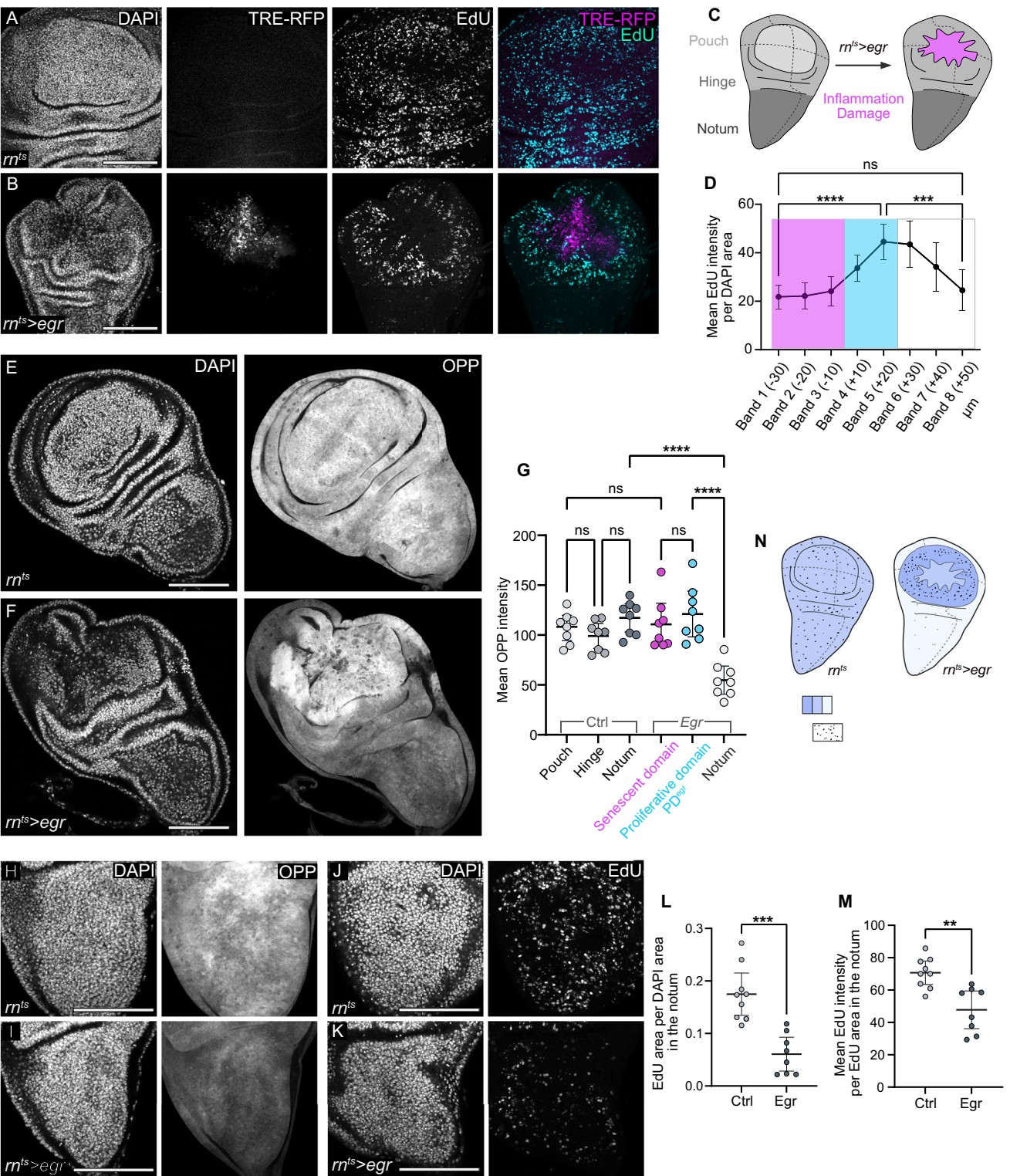

reveal a widespread decline in cell proliferation and protein synthesis in *egr*-expressing larvae, which correlates with restricted Insulin production and signalling (Fig. 2U). These observations are consistent with previous reports of reduced disc sizes following other types of tissue damage and resemble systemic changes induced by inflammatory tumours[40,42,44,45,47,48,53,62,67,74,75].

**Nutrient importers and TORC1 support regenerative growth**
How can the proliferative domain maintain high rates of protein translation and proliferation in a systemic environment that does not rely on Insulin signalling? In this environment, proliferating cells must solve two problems: (1) They must obtain and take up the right nutrients, and (2) they must drive protein translation and growth under Insulin-limiting conditions. To address the first question, we investigated if *egr*-driven Insulin restriction may induce nutrient release from the fat body, the largest nutrient storage organ in larvae. Previous studies have shown that fat body breakdown provides nutrients for tumour growth, thereby revealing an ancient programme of nutrient supply under inflammatory conditions[44,45,47,48,52,53,76]. To understand if similar catabolic changes may be induced in *egr*-expressing larvae, we

**Fig. 1 | Spatial organisation of cell proliferation and protein synthesis induced by inflammatory damage in wing imaginal disc. A, B** Control (**A**) and genetically ablated wing disc (**B**) after 24 h of *egr*-expression in the wing pouch (see Fig. 1C) under control of the *rn*-GAL4 driver (*rn*[ts] and *rn*[ts] > *egr*). TRE-RFP visualises JNK-pathway activity (magenta/grey), EdU visualises DNA replication (cyan/grey), and DAPI visualises nuclei. **C** Schematic of a third instar wing imaginal disc with pouch, notum, and hinge (left) and after 24 h of *egr*-expression in the pouch (right). Magenta identifies the JNK-signalling domain representing inflammatory damage. **D** Quantification of mean EdU intensity in the DAPI area contained within 10 μm bands segmented inward and outward from the edge of the high JNK domain in *egr*-expressing discs. Magenta = high JNK-signalling domain, cyan = domain of proliferative regeneration. Mean and 95% CI (confidence interval) shown, One-way ANOVA followed by Tukey's post-hoc test for multiple comparisons (*n* = 7 discs). *P*-values: Band 1 vs Band 5 = 0.0001; Band 5 vs Band 8 = 0.0004; Band 1 vs Band 8 = 0.9984 (ns). **E, F** Protein synthesis visualised by O-proparyl-puromycin (OPP) incorporation into newly synthesised proteins in control wing disc (**E**) and *egr*-expressing disc (**F**). **G** Mean OPP intensity in three different regions of control (pouch, hinge, and notum) and *egr*-expressing discs (high JNK signalling domain - approximating the central pouch, the proliferating region - approximating the peripheral pouch and hinge, and the notum. Mean and 95% CI are shown. One-way ANOVA followed by Tukey's post-hoc test for multiple comparisons (control: *n* = 8, *egr*-expressing disc: *n* = 8). *P*-values: Proliferative region vs Quiescent notum < 0.0001; Notum vs Quiescent notum < 0.0001; Pouch vs JNK domain = 0.9992 (ns). **H, I** OPP visualises protein synthesis in nota of control (**H**) and *egr*-expressing discs (**I**). **J, K** EdU visualise DNA replication in nota of control (**J**) and *egr*-expressing discs (**K**). **L** EdU area per DAPI area in the notum of control and *egr*-expressing discs, approximating percentage cells undergoing DNA replication. Mean and 95% CI, two-tailed Unpaired *t* test (control: *n* = 9, *egr*-expressing disc: *n* = 8), with a *p*-value = 0.0001. **M** Mean EdU intensity per EdU-positive area in the notum of control and *egr*-expressing discs, approximating relative DNA replication speed. Mean and 95% CI, two-tailed Unpaired *t*-test (control: *n* = 9, *egr*-expressing disc: *n* = 8), with a *p*-value = 0.0011. **N** Schematic of protein synthesis rates (blue shades) and mitotic activity (black dots) in control (left) and *egr*-expressing discs (right). Scale bars: 100 μm. and DAPI visualises nuclei. Fluorescence intensities reported as arbitrary units. Source data in graphs are provided as a Source Data file.

employed Nile Red staining to visualise lipid droplets within the fat body. We observed morphological changes, such as increased droplet size and 'roundness' (defined as the relationship between area and length of major axis), consistent with molecular changes associated with lipid mobilisation (Fig. 3A–C and Supplementary Fig. S3.1A)[44,48,50,77,78]. Correspondingly, fat bodies of these larvae exhibited a significant decrease in triglyceride content, similarly to fat bodies from starved larvae (Fig. 3D, E). Moreover, we find that levels of total and activated p-Akt in fat body from *egr*-expressing larvae were reduced, similar to insulin restriction observed in starved larvae (Supplementary Fig. S3.1B). In addition, levels of nuclear dFOXO-GFP were elevated (Supplementary Fig. S3.1C–E), supporting the notion that *egr*-induced insulin restriction reduces systemic Insulin signalling and promotes a catabolic fat body state.

Notably, JNK-signalling cells in *egr*-expressing discs also express *ImpL2*, *upd2* and *upd3*, factors previously implicated in fat body breakdown during tumour growth in larval and adult hosts (Fig. 3F–H and Supplementary Fig. S3.1 F–H, S3.2)[44,47,49,79]. When we ectopically expressed ImpL2 for 24 h using the wing-pouch-specific driver *rn*-GAL4, we found that this was sufficient to induce lipid droplet changes consistent with fat body catabolism (Fig. 3I–K and Supplementary Fig. S3.1 I). Moreover, *egr*-expressing larvae display activation of JAK/STAT signalling in fat body, which has also been associated with fat body catabolism (Supplementary Fig. S3.1J–L)[47,80,81]. Of note, we did not observe alterations in muscle morphology or evidence of autophagy in imaginal discs, suggesting that these tissues do not serve as a primary source for nutrients after just 24 h of *egr*-expression, different to what is observed in the chronic presence of tumours (Supplementary Fig. S3.1M–P)[44,45,47,48,52,53,76]. Combined, we conclude that the combinatorial expression of systemically acting cytokines from JNK-signalling cells in *egr*-expressing discs drives detectable nutrient mobilisation from the fat body via multiple pathways.

To determine if the observed break-down of the fat body could also generate amino acid compounds supporting the high rates of protein translation in the proliferative domain, we performed an untargeted metabolomic analysis of the hemolymph from control and *egr*-expressing larvae. Our results revealed a shift in the composition of amino acids in *egr*-expressing larval hemolymph. We observed an enrichment of many amino acids, including Leucine, Arginine or Glutamine, which are reported to activate mTORC1[82,83]. Surprisingly, many come enriched in form of mixed dipeptide species. Furthermore, elevated levels of dipeptide species and free amino acids associated with glutamate metabolism were observed, including increased concentrations of glutamine, glutamate, and glutamyl dipeptides (Fig. 3L). Notably, a signature of arginine metabolism emerged, characterised by increased levels of ornithine, N-acetyl-arginine, and aspartate alongside decreased levels of argininosuccinic acid, citrulline, and N-acetyl-ornithine. This altered metabolite profile in the hemolymph represents a flexible supply of building blocks not only for protein synthesis, but also for energy production, co-factor generation, and glutathione-based redox metabolism[84–87].

We next examined the ability of proliferating cells to absorb these amino acids. We investigated the expression of solute carrier (SLC) transporters and found that several transcripts were expressed and elevated in *egr*-expressing discs. For instance, *CG15279* (a putative glycine and proline transporter from the SLC6 family), *path* (a potential alanine and glycine transporter from the SLC36 family), *mnd* (an amino acid/polyamine transporter involved in leucine import), and *CG5535* (a putative lysine, arginine, and ornithine transporter from the SLC7 family) were all elevated (Supplementary Fig. S3.2). Upregulation of CG5535 was confirmed by immunofluorescence, and we similarly detected increased levels of CG1139 (Arcus), another SLC36 family member that may transport alanine, glycine, and proline (Fig. 3M–O and Supplementary Fig. S3.1Q, R). In addition, a potential sugar transporter, *CG3168* of the SLC22 family, and TRET-1, which is predicted to import trehalose, were strongly upregulated (Supplementary Fig. S3.1 S–U). These findings suggest that proliferating cells have an enhanced capacity for importing both amino acids and energy sources.

One regulatory branch promoting protein translation in response to amino acid availability is mediated by mTORC1[82,83,88–90]. Moreover, mTORC1 activity itself is promoted by sugar import, boosting ATP production, and hence inhibiting the mTORC1 antagonist AMPK[88–90]. The observed upregulation of amino acid and sugar transporters in *egr*-expressing discs could therefore facilitate mTORC1 activation. To demonstrate that the mTORC1 pathway supports protein synthesis during wing disc regeneration, we inhibited mTORC1 activity by feeding *egr*-expressing larvae the mTOR inhibitor Rapamycin for 24 h during *egr*-expression[15,91]. This treatment resulted in a pronounced reduction in protein synthesis within the proliferative domain of *egr*-expressing discs, demonstrating that mTORC1 activity is essential for protein translation during regeneration, as in wild-type control discs (Fig. 3P–R and Supplementary Fig. S3.1V, W). Overall, our data support a model in which fat body catabolism, enhanced amino acid uptake and mTORC1 activation together support the rapid growth and proliferation of the regenerating domain (Supplementary Fig. 3S).

## Pdk1 is upregulated in the proliferating domain

S6K, an important effector of mTORC1, directly activates protein translation[15,90]. S6K transcripts are upregulated in the wound-associated cell populations of *egr*-expressing discs (Supplementary Fig. S3.2D)[25], suggesting that S6K is positively regulated during tissue

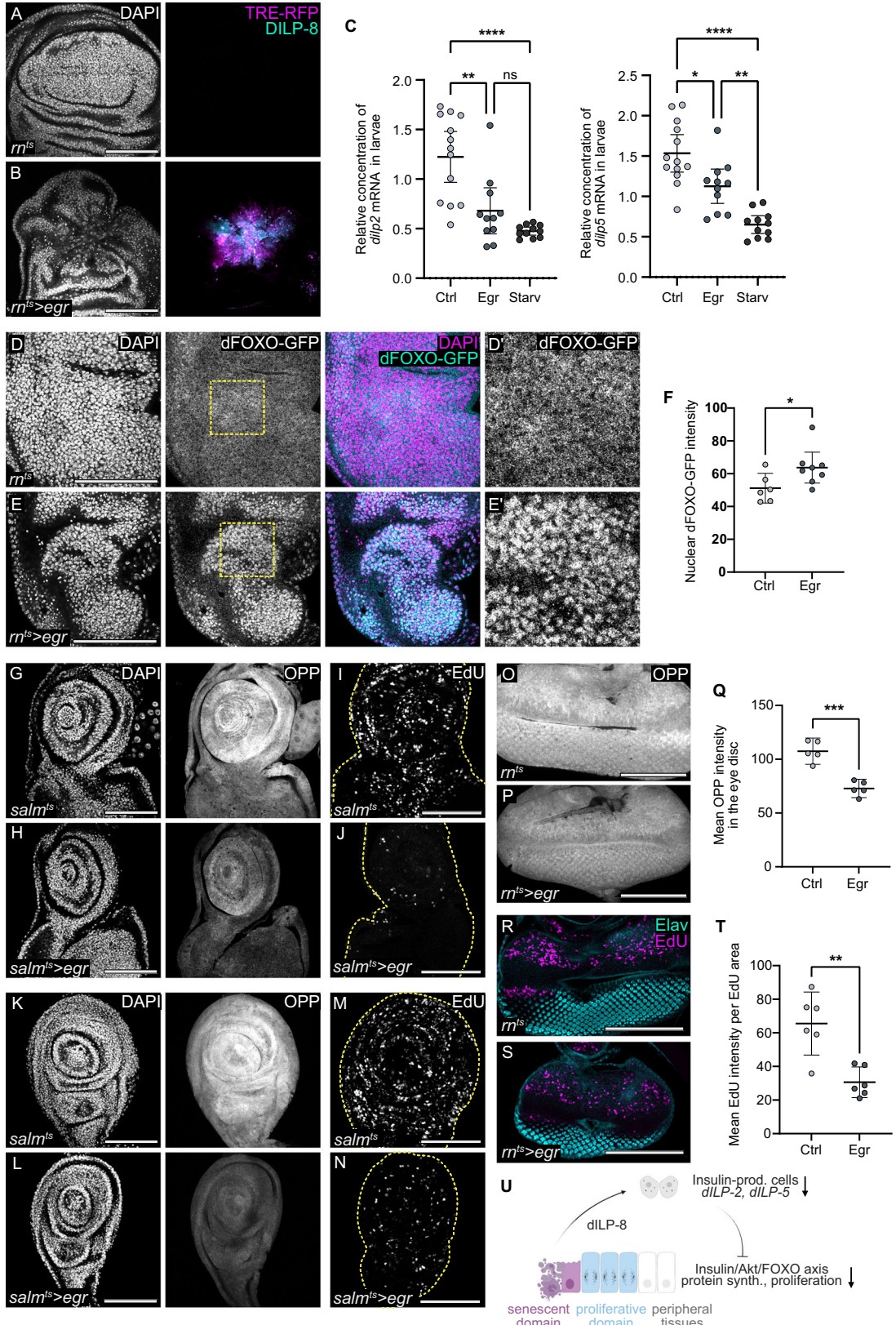

repair. Importantly, staining for phosphorylated ribosomal S6 (p-S6), a direct target of S6K, revealed that the proliferating domain of *egr*-expressing discs exhibited strongly elevated p-S6 levels (Fig. 4A–C). However, elevated p-S6 levels cannot be explained solely by mTORC1 activation of S6K because S6K requires co-activation by the Pdk1, a kinase central to canonical Insulin/PI3K signalling[14–16].

However, how can proliferating cells promote activation of S6K by Pdk1, considering that *egr*-expressing larvae exhibit insulin restriction? To understand how Pdk1 may be activated, we examined if canonical Insulin/PI3K signalling was active in the proliferating domain of *egr*-expressing discs. We therefore visualised the nuclear localisation of dFOXO, the most downstream effector of Insulin/PI3K activity, using

**Fig. 2 | Inflammatory tissue damage induces Insulin restriction and a systemic reduction in growth and cell proliferation. A, B** Dilp8-GFP (cyan) in control (**A**) and *egr*-expressing discs (**B**). TRE-RFP visualises JNK-pathway activity (magenta). dFOXO-GFP visualised using BDSC 38644. **C** Relative *dilp2* and *dilp5* mRNA in control, *egr*-expressing, and starved male larvae. *Egr*-expression was induced for 24 h in the wing pouch using *rn*-GAL4, while starved larvae underwent 24 h starvation. Mean and 95% CI, Kruskal-Wallis test followed by Dunn's multiple comparison (control larvae: $n = 13$, *egr*-expressing larvae: $n = 11$, starved larvae: $n = 11$). *P*-values for *dilp2*: Control vs egr = 0.0075; Control vs Starved < 0.0001; egr vs Starved = 0.1138 (ns). P-values for *dilp5*: Control vs egr = 0.0440; Control vs Starved < 0.0001; egr vs Starved = 0.0073. **D, E** dFOXO-GFP (cyan/grey, BDSC: 38644) in notum of control (**D**) and *egr*-expressing wing discs (**E**). DAPI (magenta/grey) visualises nuclei. Nuclear dFOXO indicates low insulin signalling. **D′** and **E′** show magnified boxed regions. **F** Quantification of mean nuclear dFOXO-GFP intensity in notum of control (**D**) and *egr*-expressing wing discs (**E**). Mean and 95% CI, two-tailed Unpaired *t* test, *p*-value = 0.0427 (control: $n = 6$, *egr*-expressing disc: $n = 8$). **G, H** OPP incorporation visualised protein synthesis in eye-antennal disc, dissected from larvae with control (**G**) or *egr*-expressing (**H**) wing discs using *salm*-GAL4 expressed in the central pouch. **I, J** EdU incorporation visualises DNA replication in eye-antennal discs, dissected from larvae with control (**I**) or *egr*-expressing (**J**) wing discs using *salm*-GAL4 driver. The yellow line outlines the disc. **K, L** OPP incorporation visualised protein synthesis in the leg disc, dissected from larvae with

control (**K**) or *egr*-expressing (**L**) wing discs using *salm*-GAL4. **M, N** EdU incorporation visualises DNA replication in leg imaginal discs, dissected from larvae with control (**M**) or *egr*-expressing (**N**) wing discs using *salm*-GAL4. The yellow line outlines the disc. **O, P** OPP incorporation visualised protein synthesis in the eye disc, dissected from larvae with control (**O**) or *egr*-expressing (**P**) wing discs using *rn*-GAL4. **Q** Mean OPP intensity in eye discs, dissected from larvae with control or *egr*-expressing wing discs. Mean and 95% CI, two-tailed Unpaired *t* test, *p*-value = 0.0002 (control: $n = 5$, *egr*-expressing disc: $n = 5$). **R, S** EdU visualises DNA replication (magenta) in eye discs, dissected from larvae with control (**R**) or *egr*-expressing (**S**) wing discs using *rn*-GAL4. Photoreceptors are marked by Elav (cyan). **T** Mean EdU intensity per EdU-positive area (relative DNA replication speed), in eye discs dissected from larvae with control or *egr*-expressing wing discs. Mean and 95% CI, two-tailed Unpaired *t* test, *p*-value = 0.0015 (control: $n = 6$, *egr*-expressing disc: $n = 6$). **U** Model: dILP-8 from senescent-like cells in the high JNK domain causes downregulation of dILP-2 and dILP-5 from insulin-producing cells (IPCs) in the larval brain, reducing systemic Insulin/Akt/FOXO signalling, protein synthesis and proliferation. Scale bars: 100 μm. and DAPI visualises nuclei. Fluorescence intensities reported as arbitrary units. Source data in graphs are provided as a Source Data file. Experiments depicted in 2G-N were carried out twice with $n > 6$ discs for control or experiment. Illustrations were created in Biorender Classen, A. (2025) https://BioRender.com/h63vwwi.

two independent GFP-tagged lines. We observed high nuclear levels of dFOXO in the proliferating domain of *egr*-expressing discs, suggesting that Insulin/PI3K signalling there is low, which is consistent with our evidence for systemic Insulin restriction (Fig. 4D–G and Supplementary Fig. S4A–C). This conclusion is further supported by unchanged InR levels and even reduced levels of Akt and phosphorylated Akt (P-Akt) in the proliferative domain (Supplementary Fig. S4D–K). In contrast, we found that Pdk1 expression is strongly upregulated in the proliferative domain (Fig. 4H–J) (see Materials & Methods, section *Drosophila* genetics). Similar upregulation of Pdk1 was also observed in other models of tissue damage and regeneration, including those using expression of *hid* or *reaper* to drive cell death. In these models, elevated Pdk1 levels also correlated with high protein translation, demarcating the proliferating domains (Fig. 4K, L and Supplementary Fig. S4L–Q). In fact, we found that high levels of Pdk1 correlated with high levels of p-S6 staining in *egr*-expressing discs (Fig. 4M, N). Collectively, these observations suggest that Pdk1 levels may play a central role in regenerative proliferation.

### Pdk1 upregulation is sufficient to drive growth and is necessary for regenerative proliferation

To understand if Pdk1 upregulation is a central mechanism driving regenerative proliferation, we expressed either a wild-type or a constitutively active form of Pdk1 in the wing pouch for 24 h using *rn-GAL4*. Compared to control discs, both forms of Pdk1 resulted in higher levels of protein translation and larger pouch sizes. Larger pouch sizes correlated with more cell proliferation, as EdU incorporation in Pdk1 expressing domains was elevated (Fig. 5A–F and Supplementary Fig. S5A–D). These results demonstrate that high Pdk1 levels are sufficient to support a metabolic programme characteristic of regenerative proliferation. Notably, the fact that overexpression of a wild-type Pdk1 alone enhances protein synthesis and proliferation suggests that Pdk1 activity scales with expression levels, functioning independently of other canonical upstream signalling inputs, such as Insulin/PI3K. Of note, Pdk1 can auto-activate via transphosphorylation when recruited to the plasma membrane[92,93], reflecting potentially Insulin-independent means for Pdk1 activation.

To provide evidence that Pdk1 function is also necessary to support proliferation in *egr*-expressing discs, we genetically reduced Pdk1 function by establishing heterozygosity for a null allele of *Pdk1*[94]. In control discs, this heterozygosity did not affect overall protein synthesis compared to wild-type discs, indicating that under normal developmental conditions, a single copy of *Pdk1* is sufficient for growth. However, protein synthesis in the proliferating domain of

*egr*-expressing discs heterozygous for *Pdk1* was significantly reduced (Fig. 5G–K). This finding demonstrates that the proliferating domain is particularly sensitive to reductions in Pdk1 levels and underscores the necessity for elevated Pdk1 to support protein synthesis during regeneration. To provide further proof, we expressed a validated RNAi against *Pdk1* specifically in proliferating cells of *egr*-expressing discs under control of the DUAL Control genetic ablation system (Supplementary Fig. S5E–H)[65]. Consistent with our heterozygous mutant results, *Pdk1* knockdown led to a marked decrease in protein translation in the proliferating domain, confirming a cell type specific function for Pdk1 upregulation in the proliferating domain during regeneration (Fig. 5L–N). Overall, these findings establish that Pdk1 upregulation is not only sufficient to drive protein translation and growth but is also critically required to sustain protein translation in the proliferative domain of regenerating imaginal discs under conditions of systemic insulin restriction (Fig. 5O).

### Pdk1 is regulated by JAK/STAT signalling

To understand how Pdk1 levels are controlled during regeneration, we examined the role of JAK/STAT signalling, which is activated in the proliferating domain of *egr*-expressing discs to promote cell proliferation and survival (Fig. 6A, B)[24–26,32,64,65]. We therefore expressed STAT92E in the posterior compartment of wing imaginal discs, which elevates JAK/STAT signalling, using the *en-GAL4* driver[26]. This manipulation caused an increase in levels of Pdk1-GFP, phospho-S6 and OPP incorporation (Fig. 6C–H and Supplementary Fig. S6C–E), demonstrating that JAK/STAT activation is sufficient to promote Pdk1 upregulation, S6 activation and protein translation. Conversely, when we expressed an RNAi construct to knock down *STAT92E* in the posterior compartment during normal development, we observed a decrease in both Pdk1-GFP levels and OPP incorporation, suggesting that JAK/STAT activity can limit Pdk1 levels and protein translation (Fig. 6I–L). Importantly, the JAK/STAT-Pdk1-S6 axis is also essential for regeneration: we genetically reduced *STAT92E* function in *egr*-expressing discs by establishing heterozygosity for a null allele of *STAT92E* and found that it reduced OPP incorporation specifically in *egr*-expressing discs but not in control discs (Fig. 6M–O and Supplementary Fig. S6F, G). Moreover, when we expressed an RNAi against *STAT92E* specifically in proliferating cells of *egr*-expressing discs under control of the DUAL Control genetic ablation system[65], the elevated protein translation was abrogated (Fig. 6P–R and Supplementary Fig. S6H, I). Combined, these findings reveal a function for JAK/STAT signalling in tissue repair and regeneration by promoting Pdk1 upregulation, which, together with

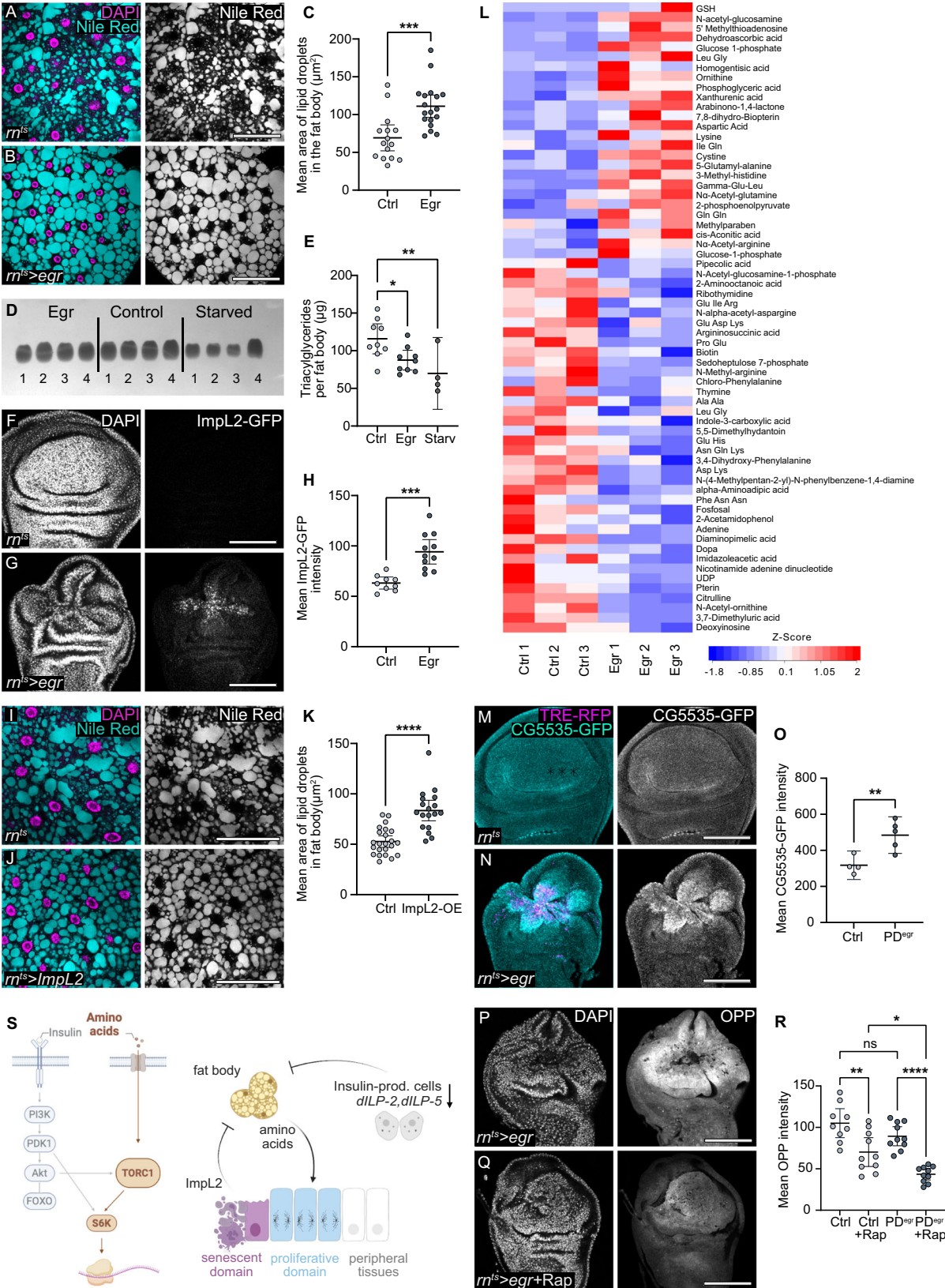

mTORC1, supports insulin-independent growth in the proliferating domain (Supplementary Fig. S6J).

## JAK/STAT-Pdk-1-S6K signalling is linked to tumour growth

We wanted to understand if the JAK/STAT-Pdk1-S6 signalling axis observed during regenerative proliferation also plays a role in tumour growth, given that tumours often co-opt physiological repair processes for their pathological purposes. We thus analysed imaginal discs in which tumorigenesis was induced by both reducing the function of the tumour-suppressor gene *scrib*[95] and ectopically expressing oncogenic *Ras*[V12][96,97]. We used the *rn-GAL4* driver to drive expression of *scrib-RNAi* and *Ras*[V12] in the entire wing pouch, which causes

**Fig. 3 | Fat body catabolism and nutrient release to hemolymph facilitates regenerative proliferation through mTORC signalling pathway. A, B** Nile Red staining (cyan/grey) of fat body dissected from larvae with control (**A**) or *egr*-expressing (**B**) wing discs. DAPI (magenta) visualises nuclei. **C** Mean area of lipid droplets in fat body dissected from larvae with control (**A**) or *egr*-expressing (**B**) wing discs. Mean and 95% CI, two-tailed Unpaired *t* test. *p*-value = 0.0004. (control: *n* = 15, *egr*-expression in discs: n = 18). **D, E** Triacylglycerides (TAG) levels in fat bodies dissected from larvae with *egr*-expressing wing disc, control or starved (16 h) larvae. Mean and 95% CI, One-way ANOVA followed by Dunnett's test for multiple comparison (control larvae: *n* = 9, *egr*-expressing larvae: *n* = 9 and starved larvae: *n* = 4). *p*-values: control vs egr = 0.0353; control vs starved =0.0076. **F, G** ImpL2-GFP in control (**A**) and *egr*-expressing discs (**B**). DAPI visualises nuclei. **H** Mean ImpL2-GFP intensity in control (**F**) and *egr*-expressing discs (**G**). Mean and 95% CI, two-tailed Welch's *t* test, *p*-value = 0.0001 (control: *n* = 9, *egr*-expressing discs: *n* = 11). **I, J** Nile Red staining of fat body (cyan/grey) from larvae with control (**I**) or ImpL2-expressing wing discs using *rn*-GAL4 (**J**). **K** Lipid droplet areas in fat body from larvae with either control or ImpL2-expressing wing discs using *rn*-GAL4 (24 h). Mean and 95% CI, two-tailed Mann-Whitney test, *p*-value < 0.0001 (control: *n* = 23, experiment: *n* = 18). **L** Heat map showing relative changes of metabolite concentrations in the larval hemolymph from control and *egr*-expressing larvae. Data were quantile normalised and analysed using a two-sided, unpaired Wilcoxon rank-sum test. Metabolites with at least < 0.75 and > 1.5-fold change were selected. Metabolites were ordered by log2 fold changes shown as Z-scores. Sample size *n* = 3 per condition. **M, N** CG5535-GFP in control (cyan/grey) (**M**) and *egr*-expressing discs (**N**). TRE-RFP visualises JNK- activity (magenta). **O** Mean CG5535-GFP intensity in the pouch of control and the proliferative domain of *egr*-expressing discs (PD^egr^). Mean and 95% CI, two-tailed Unpaired *t* test, *p*-value = 0.0092 (control: *n* = 4, *egr*-expressing discs: *n* = 5). **P, Q** OPP visualises protein synthesis in *egr*-expressing wing discs from larvae fed on food without (**P**) or with rapamycin (200 μM) (**Q**) for 24 h during *egr*-expression. **R** Mean OPP intensity in the pouch of control and the proliferative domain of *egr*-expressing discs (PD^egr^) from larvae fed on food without or with rapamycin. Mean and 95% CI, One-way ANOVA followed by Tukey's post-hoc test for multiple comparisons (control: non-fed: *n* = 9, fed: *n* = 10; *egr*-expressing disc: non-fed: *n* = 10, fed: *n* = 10). PD^egr^ vs PD^egr^ + Rapamycin: *p* < 0.0001, Control + Rapamycin vs PD^egr^ + Rapamycin: *p* = 0.0167 and Control vs Control + Rapamycin: *p* = 0.0017. **S** Model: ImpL2 from senescent-like cells in high JNK domain represses fat body anabolism via inducing insulin resistance, promoting nutrient mobilisation. This branch may reinforce the metabolic switch induced by Insulin restriction. Released nutrients (e.g., amino acids) enter hemolymph and support mTORC1/S6K activation in the proliferative domain. Scale bars: 100 μm. Fluorescence intensities are reported as arbitrary units. Source data in graphs are provided as a Source Data file. Illustrations were created in Biorender Classen, A. (2025) https://BioRender.com/h63vwwi.

pronounced overproliferation[26,51,53]. This tumour model is characterised by the coordinated but spatially separated activation of JNK and JAK/STAT signalling networks, whose regulatory and functional characteristics resemble those active during tissue repair in *egr*-expressing discs (Supplementary Fig. S7A)[26]. In *Ras^V12^,scrib-RNAi* expressing discs, we found a striking correlation between regions of JAK/STAT activation, elevated protein translation and increased Pdk1 level (Fig. 7A–F). An independent tumour model of *Psc-Su(z)2* tumours also shows correlated elevation of JAK/STAT activation and protein synthesis (Supplementary Fig. S7B, C). Importantly, these tumours, like *egr*-expressing discs, induce all hallmarks of systemic cachexia, including reduced protein translation in the non-transformed notum (Fig. 7G–I), and must therefore overcome growth restrictions imposed on peripheral tissues[40,53,76]. Our observation demonstrates that tumours likely coopt JAK/STAT activation and Pdk1 elevation to support protein translation and thus tumour proliferation.

### Pdk1 reduction in peripheral discs during growth restriction

Strikingly, Pdk1 levels appear to be a target of systemic regulation during Insulin restriction in imaginal discs. In *egr*-expressing larvae, we consistently observed that Pdk1 levels were significantly lower in peripheral imaginal disc, including the notum, leg and eye, if compared to wild type control discs (Fig. 8A–F and Supplementary Fig. S8). This observation was not limited to *egr*-expressing larvae; even in the peripheral imaginal discs of *Ras^V12^,scrib-RNAi* tumour-bearing larvae, Pdk1 levels were markedly reduced (Fig. 8G–I). This targeted reduction of Pdk1 in peripheral imaginal discs is consistent with the overall decrease in protein synthesis and growth observed in these regions. Such downregulation suggests that systemic signals, maybe even Insulin restriction, actively suppress Pdk1 expression in non-regenerative areas. In contrast, JAK/STAT signalling in regenerative areas can counteract Pdk1 suppression and locally elevate Pdk1 levels and function. This dual regulation -upregulation in regenerative zones and downregulation in non-essential areas -highlights Pdk1's central role in mediating the balance between local regenerative demands and systemic metabolic reprogramming. Consequently, Pdk1 emerges not only as a key driver of insulin-independent regenerative proliferation but also as an integrative node that coordinates systemic growth control during tissue repair and tumorigenesis.

## Discussion

Our study reveals how signalling through a JAK/STAT-Pdk1-S6K axis promotes metabolic reprogramming to drive tissue repair and regeneration in an environment of systemic growth restriction (Fig. 8J, K). We demonstrate that in response to inflammatory damage, a senescent subpopulation of cells at the centre of tissue damage induces insulin restriction via action of dILP8 on IPCs, as well as insulin resistance via action of ImpL2 on the fat body. The resulting systemic reduction of insulin signalling is evident from the decreased expression of *dILP2* and *dILP5* in IPCs, as well as the enhanced nuclear localisation of dFOXO, low rates of protein synthesis and proliferation in peripheral tissues, or reduced Akt signalling and nutrient mobilisation in the fat body. These findings align with previous work suggesting that systemic growth restriction, for example via repression of Ecdysone signalling, helps synchronise regeneration with developmental timing and prevents the overgrowth of undamaged tissues[41,42,57,58,98].

However, slowing down peripheral growth poses a challenge to the regenerating cell population, which must continue to support high anabolic activity despite reduced canonical Insulin/PI3K/AKT signalling. The regenerating cell population circumvents this limitation by engaging a previously uncharacterised mechanism to sustain protein synthesis. Specifically, we find that Pdk1 is robustly upregulated in the proliferative domain of *egr*-expressing discs, and this upregulation is sufficient and necessary to drive ribosomal S6 phosphorylation and activate protein translation, even in the context of low systemic Insulin signalling. The capacity of Pdk1 to function independently of insulin positions Pdk1 as a central regulator of regenerative metabolism. Notably, our results mirror earlier observations of AKT-independent, yet Pdk1-dependent, growth during *Drosophila* development[94,99]. During regeneration, we identify JAK/STAT as an upstream regulator of Pdk1, and consequently as a regulator for phosphorylation of ribosomal S6 protein and activation of protein synthesis. Our work thus uncovers a pathway whereby JAK/STAT signalling can mediate the known proliferation-promoting function in imaginal disc development, regeneration, and tumour growth[30]. Of note, the increase in Pdk1 protein levels likely arises from JAK/STAT regulated post-translational mechanisms affecting Pdk1 stability, as we did not observe an increase in *Pdk1* transcripts in *egr*-expressing discs[25]. Such post-translational regulation of Pdk1 may also underlie the differential regulation of Pdk1 between regenerative and peripheral tissues. While we find that the proliferative domain actively upregulates Pdk1 to overcome insulin restrictions, we also find that the peripheral imaginal discs downregulate Pdk1. Based on our experiments demonstrating that Pdk1 levels are limiting for protein translation, the peripheral reduction in Pdk1 may contribute to the reduction in protein synthesis and growth

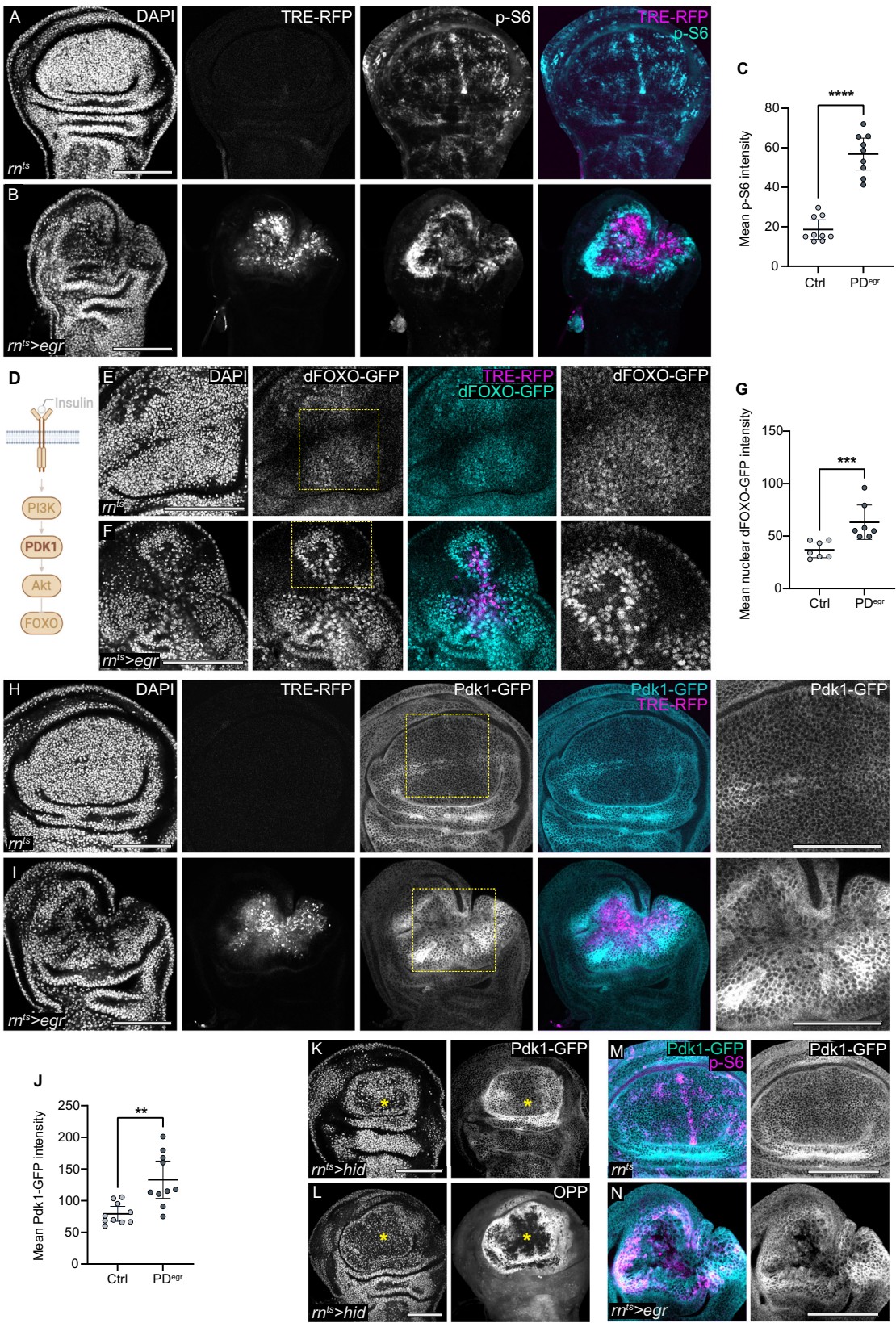

of other imaginal discs. This dual regulation emphasises the surprisingly central role of Pdk1 in integrating local demands for regeneration with systemic constraints on growth.

Furthermore, our work reveals that fat body catabolism, which mobilises nutrients necessary for regeneration, is activated in *egr*-expressing larvae. In addition to the remodelling of the lipid stores, proteins may also be catabolized, as suggested by an altered amino acid signature in the hemolymph. The upregulation of amino acid and carbohydrate transporters in the proliferative domain further suggests that regenerating cells actively import these mobilised nutrients, thereby fueling mTORC1 activation and subsequent protein synthesis. Earlier studies find elevated levels of ornithine, glutamate, and

**Fig. 4 | Pdk1 is upregulated in the proliferative domain. A, B** Phosphorylated-S6 (pS6, cyan/grey) in control (**A**) and *egr*-expressing wing discs (**B**). TRE-RFP visualises JNK activity (magenta or grey). **C** Mean p-S6 intensity in the pouch of control and the proliferative domain of *egr*-expressing discs (PD^egr). Mean and 95% CI, two-tailed Mann-Whitney test, *p*-value < 0.0001 (control: *n* = 9, *egr*-expressing discs: *n* = 9). **D** Schematic of canonical Insulin/Akt/FOXO signalling. Phosphatidylinositol kinase-1 (Pdk1) is a key serine/threonine kinase in this pathway. **E, F** dFOXO-GFP (cyan/grey, BDSC: 38644) in control (**E**) and *egr*-expressing wing discs (**F**). TRE-RFP visualises JNK activity (magenta/grey). Box marks the inset region. **G** Mean nuclear dFOXO-GFP intensity in the pouch of control or proliferative domain of *egr*-expressing discs (PD^egr). Mean and 95% CI, two-tailed Mann-Whitney test, *p*-value = 0.0006 (control: *n* = 7, *egr*-expressing disc: *n* = 7). **H, I** Pdk1-GFP (cyan or

grey) in control (**H**) and *egr*-expressing discs (**I**). TRE-RFP visualises JNK activity (magenta or grey), DAPI visualises nuclei. Box marks the inset region. **J.** Mean Pdk1-GFP intensity in the pouch of control or proliferative domain of *egr*-expressing discs (PD^egr). Mean and 95% CI, Two-tailed Welch's *t* test, *p*-value = 0.0025 (control: *n* = 10, *egr*-expressing disc: *n* = 10). **K, L** Pdk1-GFP (**K**) and OPP (**L**) in *hid*-expressing discs. an asterisk marks the damaged region. **M, N** Pdk1-GFP (cyan/grey) and p-S6 (magenta) in control (**M**) and *egr*-expressing wing discs (**N**). Scale bars: 100 µm. DAPI visualises nuclei. Fluorescence intensities are reported as arbitrary units. Source data in graphs are provided as a Source Data file. Experiments depicted in K-N were carried out twice with *n* > 6 discs for control or experiment. Illustrations were created in Biorender Classen, A. (2025) https://BioRender.com/h63vwwi.

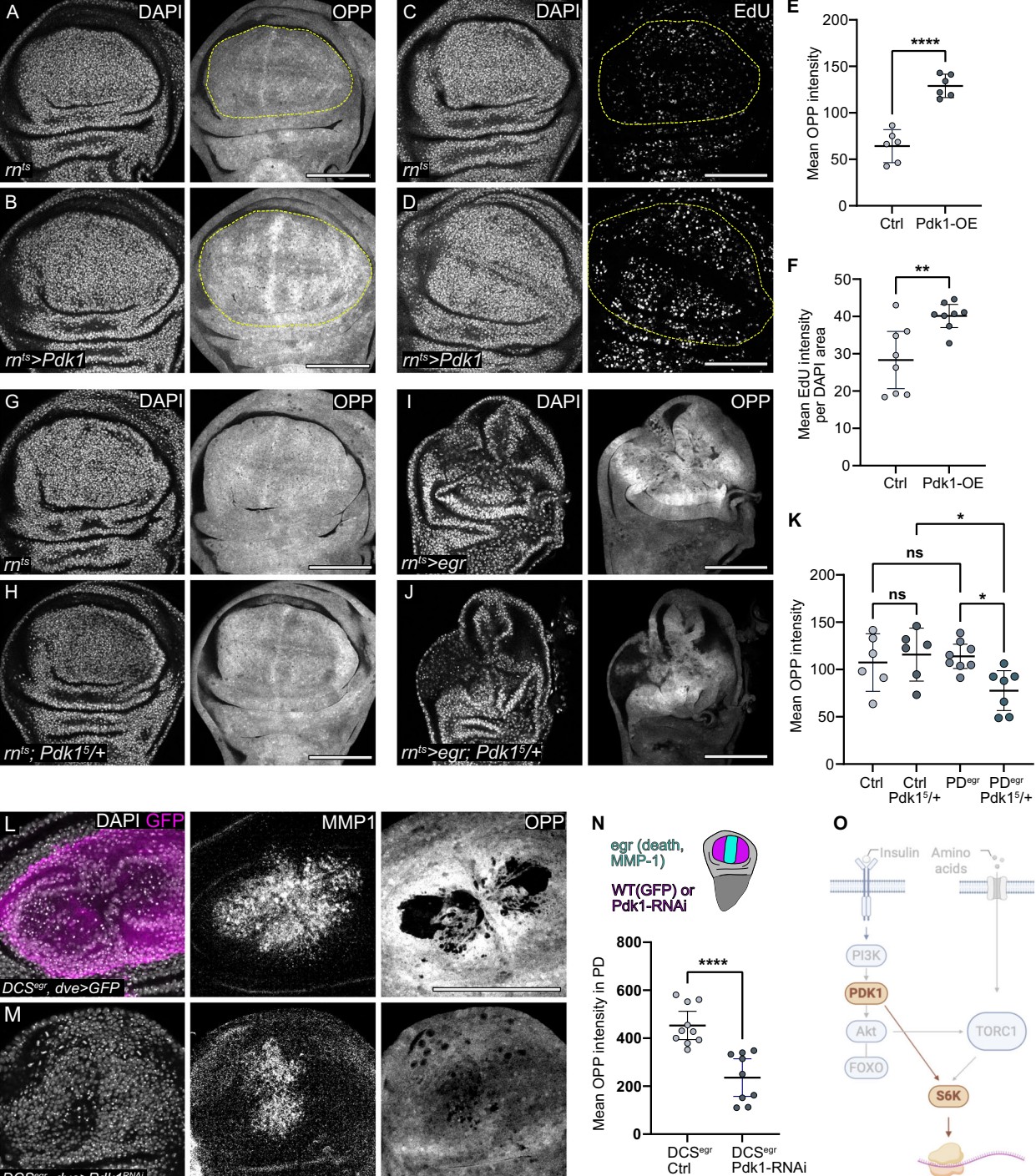

**Fig. 5 | Pdk1 upregulation is sufficient and necessary for regenerative growth and proliferation. A, B** OPP incorporation in control (**A**) and wing discs expressing Pdk1 for 24 h using *rn*-GAL4 (**B**). DAPI visualises nuclei. Line indicates boundary of pouch based on landmark folds, approximating the *rn*-GAL4 domain. **C, D** EdU incorporation in control (**C**) and discs expressing UAS-Pdk1 for 24 using *rn*-GAL4 (**D**). The line indicates the boundary of the rn-GAL4 expression domain. **E**. Mean OPP intensity in the pouch of control and *Pdk1*-expressing wing discs. Mean and 95% CI, two-tailed Unpaired *t* test, *p*-value < 0.0001 (control: *n* = 6, *Pdk1*-expressing disc: *n* = 6). **F** Mean EdU intensity per DAPI area as proxy for relative DNA replication speed, in control and *Pdk1*-expressing pouches. Mean and 95% CI, two-tailed Welch's *t* test, *p*-value = 0.0078 (control: *n* = 8, *Pdk1*-expressing disc: *n* = 8). **G–J** OPP incorporation in control (**G, H**) and *egr*-expressing wing discs (**I, J**), either wild type (**G, I**) or heterozygous mutant for the $Pdk1^5$ null allele (**H, J**). **K** Mean OPP intensity in the pouch of control or proliferative domain of *egr*-expressing discs (PD$^{egr}$), either wild type or heterozygous mutant for $Pdk1^5$ null allele. Mean and 95% CI, One-way ANOVA followed by Tukey's post-hoc test for multiple comparisons. (control:*n* = 6, control, $Pdk1^5$/+: *n* = 6; *egr*:n = 8, *egr*, $Pdk1^5$/+: *n* = 7). PD$^{egr}$ vs PD$^{egr}$: $Pdk1^5$/+: *p*-value = 0.0300 and Control: $Pdk1^5$/+ vs PD$^{egr}$: $Pdk1^5$/+: *p*-value = 0.0350. **L, M** DUAL Control system (DCS) allows to manipulate the proliferative domain. A single heat shock activates both *egr*-expression in the *salm*-domain (tracked by pyknotic nuclei and MMP-1 upregulation) and genetic manipulation in the proliferative domain via *dve*-GAL4 (tracked by UAS-GFP co-expression in **L**). MMP1 is a target gene of JNK activated by Egr, approximating the domain of *salm>egr*. *Dve*-GAL4 expresses in the pouch and pouch fold, morphological landmarks approximating the *dve*-GAL4 domain driving expression of *Pdk1-RNAi* (**M**) for 24 h. Protein synthesis is visualised by OPP incorporation in control (**L**) and *Pdk1-RNAi* (**M**) discs. **N** Mean OPP intensity in the proliferative domain of control (*DCS*$^{egr}$, ctrl) and *Pdk1* knockdown (*DCS*$^{egr}$, *Pdk1 RNAi*) discs with *egr*-expression in the *salm*-domain. A schematic of the *egr*-expressing region (cyan, trackable by cell death and MMP-1) and the *dve*-GAL4-expressing region (magenta, trackable by UAS-GFP co-expression) used for *Pdk1* knockdown. Mean and 95% CI, two-tailed unpaired *t* test (*p* < 0.0001). (*DCSegr*, control: *n* = 10; *DCSegr*, *Pdk1 RNAi*: *n* = 9). **O** Scheme highlighting the Pdk1-branch driving S6K activation. Scale bars: 100 µm. DAPI visualises nuclei. Fluorescence intensities are reported as arbitrary units. Source data in graphs are provided as a Source Data file. Illustrations were created in Biorender Classen, A. (2025) https://BioRender.com/h63vwwi.

glutamine in rat wound fluids[100], similar to the ornithine, glutamate, and glutamine signature in hemolymph from *egr*-expressing larvae. Our observations suggest that glutamate metabolism may be important for regeneration, consistent with its described roles in tumour growth, inflammation, and defence against oxidative stress[10,84,101].

Together, our results support a model in which *egr*-induced inflammation induces systemic insulin restriction and insulin resistance, thereby limiting resource availability in peripheral tissues. At the same time, local activation of pathways such as JAK/STAT, Pdk1, or nutrient importers prioritises tissue repair at the damage site. A potential limitation of our study lies in the high levels of *eiger*-expression used in this genetic model, which may not fully reflect physiological conditions. In fact, as we previously reported, *egr*-expressing discs can resemble chronic wound states, where prolonged inflammatory signalling impairs effective tissue regeneration[26]. However, in the widely studied *Ras*$^{V12}$,*scrib-RNAi* tumour model, we observe that regions with high JAK/STAT activity also exhibit elevated Pdk1 expression and protein translation, suggesting that tumours which can induce cachexia also exploit this pathway to sustain growth despite low systemic insulin signalling. Given that Pdk1 can bypass the need for Insulin/AKT/FOXO signalling, it may not be surprising that Pdk1 upregulation is found in many cancers[102]. While oncogenic mutations may facilitate nutrient use in such a cachexic tumour environment[45,52,103,104], our work specifically reveals how wild-type tissues adapt within a physiological repair programme to overcome systemic inhibition of growth.

In summary, our study uncovers a regulatory network wherein fat body catabolism, enhanced nutrient uptake, mTORC1 activation, and the instructive JAK/STAT-Pdk1 axis converge to support regenerative proliferation in a cachexic environment. Undoubtedly, this is an ancient stress response and repair programme designed to distribute nutrients from an energy store to the site of tissue damage. These findings not only advance our understanding of tissue repair mechanisms in *Drosophila* but may also offer insights into conserved pathways in mammals, as a domain characterised by ribosomal S6 protein phosphorylation was recently described in mouse and pig skin wounds, suggesting potential similarities in regenerative programmes across species[18].

## Method

### *Drosophila* maintenance
All experiments were performed on *Drosophila melanogaster*. Fly strains (see Supplementary Table S1) were maintained on standard fly food (10 L water, 74,5 g agar, 243 g dry yeast, 580 g corn flour, 552 ml molasses, 20.7 g Nipagin, 35 ml propionic acid) at 18 °C – 22 °C. Larvae from experimental crosses were allowed to feed on Bloomington formulation (175.7 g Nutry-Fly,1100 ml water 20 g dry yeast, 1.45 g Nipagin in 15 ml Ethanol, 4.8 ml Propionic acid) and raised at 18 °C or 30 °C to control GAL80ts-dependent induction of GAL4/UAS. Our experimental design did not consider differences between sexes, unless for genetic crossing schemes.

### *Drosophila* genetics
To induce expression of UAS-constructs, such as *UAS-egr*, under the control of rn-GAL4 in the wing pouch, experiments were carried out as described in[24,28,62] with minor modifications. Briefly, larvae of genotype *rn-GAL4, tub-GAL80*$^{ts}$ and carrying the desired *UAS*-transgenes were staged with a 6 h egg collection and raised at 18 °C at a density of 50 larvae/vial. Overexpression of transgenes was induced by shifting the temperature to 30 °C for 24 h on the seventh day (D7) after egg deposition (AED) to relieve temperature-sensitive GAL80ts repression of GAL4. Larvae were dissected after 24 h of *egr*-expression. Control genotypes were generated by crossing rn-GAL4, tub-GAL80ts into a wild-type background. The DUAL Control genetic ablation system (DCS) was employed, wherein a single heat shock simultaneously activates genetic cell ablation by *egr*-expression within the *salm* domain and GAL4 expression in the *dve-GAL4* domain (*hsFLP; hs-p6S::zip, lexAOp-egrNI/CyO; salm-zip::LexA-DBD, DVE»GAL4)*[65]. Specifically, larvae were raised at 25 °C and subjected to a 1-hour heat shock at 37 °C on Day 4.5, followed by dissection 24 h post-heat shock. The Pdk1-GFP line (BDSC: 59836) was characterised using the following approaches: a complex expression pattern can be observed in wild type tissues in immunofluorescence, where GFP is detected in the cytoplasm but also at membranes (see for example, Fig. 6C). Tissues form larvae produce a band of the expected size on Western blot. Three independent RNAi lines (BDSC: 27725, BDSC: 34936, BDSC: 36071) targeting the 3' region of the Pdk1 transcript downstream of the GFP-cassette insertion site can robustly knock down GFP expression (see for example, Fig. 5E). We conclude that a full-length, membrane-recruitable and genetically tractable protein is produced from the endogenous locus. Thus, the annotation of the insertion site in FlyBase is incorrectly oriented.

### Immunohistochemistry of wing imaginal discs
Wing discs from third instar larvae were dissected and fixed for 15 minutes at room temperature in 4% paraformaldehyde in PBS. Washes were performed in PBS containing 0.1% TritonX-100 (PBT). The discs were then incubated with primary antibodies (listed in Supplementary Table S2) in 0.1% PBT, gently mixed overnight at 4 °C. During incubation with cross-absorbed secondary antibodies coupled to Alexa Fluorophores at room temperature for 2 h, tissues were counterstained with DAPI (0.25 ng/µL, Sigma, D9542). Tissues were

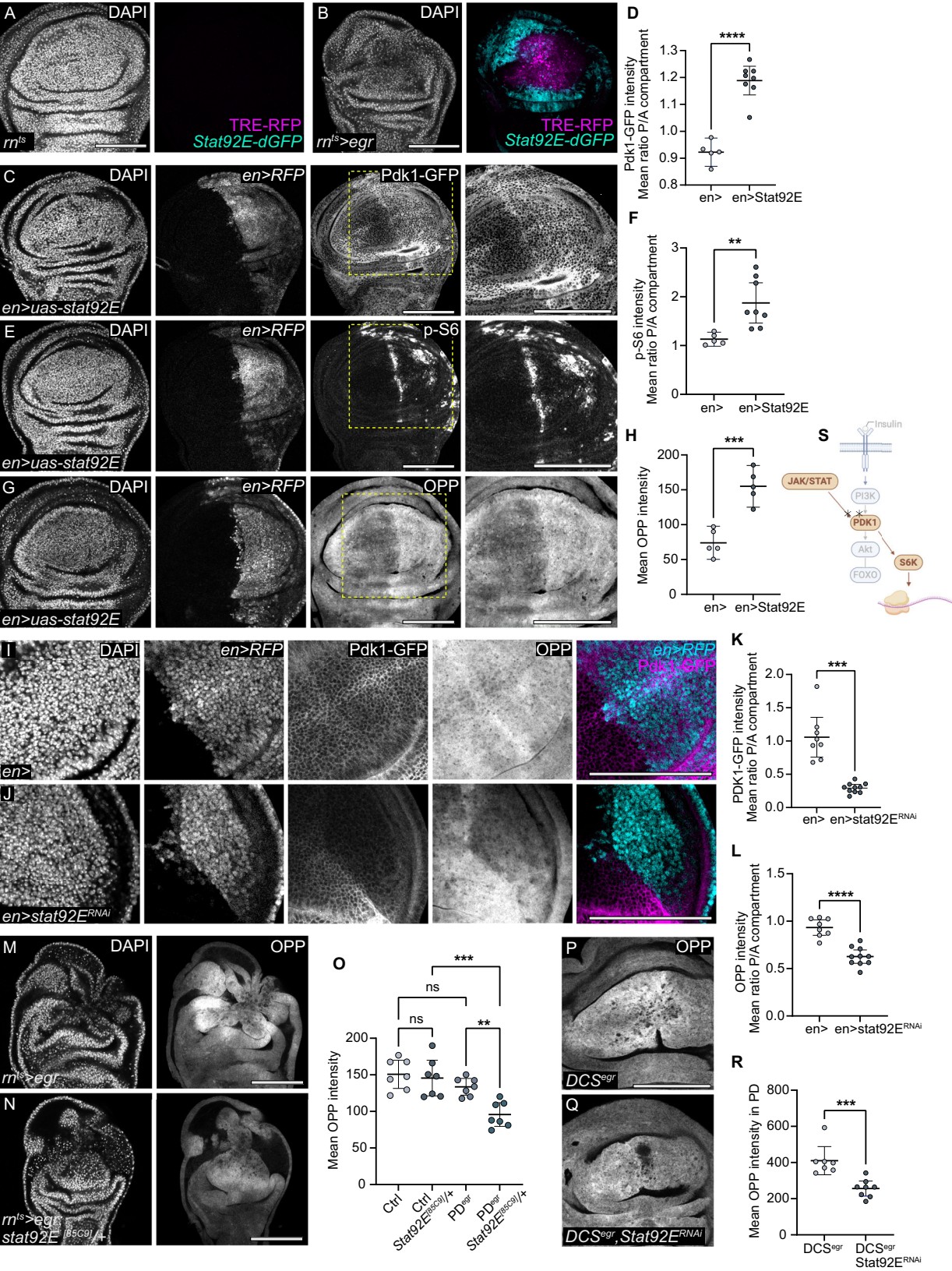

mounted using SlowFade Gold Antifade (Invitrogen, S36936). To ensure comparability in staining between different genotypes, experimental and control discs were processed in the same vial and mounted on the same slides whenever possible. Images were acquired using the Leica TCS SP8 Microscope, using the same confocal settings for linked samples and processed using tools in Fiji.

**Protein synthesis assays using OPP- Click-iT staining**

OPP Assays were performed using Click-iT® Plus OPP Protein Synthesis Assay Kits (Invitrogen Molecular Probe) according to the manufacturer's instructions. Briefly, larvae were dissected, and inverted cuticles were incubated with a 1:1000 dilution of Component A in Schneider's medium for 15 min on a nutator. Larval cuticles were fixed

**Fig. 6 | Pdk1 is regulated by JAK/STAT signalling. A, B** A control and *egr*-expressing wing disc expressing TRE-RFP (magenta) and Stat92E-dGFP (cyan) reporters. **C, E, G** Wing discs expressing *STAT92E* in the posterior compartment under *en*-GAL4, UAS-RFP. Pdk1-GFP (**C**), p-S6 (**E**), and OPP incorporation (**G**) shown. **D, F, H** Mean Pdk1-GFP (**D**), p-S6 (**F**) and OPP incorporation (**H**) levels, expressed as posterior-anterior (P/A) ratio in control and discs expressing *STAT92E* under the control *en*-GAL4. **D** two-tailed Unpaired *t* test, *p*-value < 0.0001 (control: *n* = 5, experiment: *n* = 8). **F** two-tailed Welch's *t* test, *p*-value = 0.0033 (control: *n* = 5, experiment: *n* = 8). **H** two-tailed Unpaired *t* test, *p*-value = 0.0004 (control: *n* = 5, experiment: *n* = 5). Mean and 95% CI shown. **I, J** Control wing disc (**I**) and wing discs expressing *STAT92E-RNAi* (**J**) in the posterior compartment under control of *en*-GAL4, UAS-RFP. Pdk1-GFP and OPP incorporation shown. **K** Mean Pdk1-GFP levels expressed as posterior-anterior (P/A) ratio in control and discs expressing *STAT92E-RNAi* using *en*-GAL4. Mean and 95% CI, two-tailed Welch's *t* test, *p*-value = 0.0004 (control: *n* = 8, experiment: *n* = 10). **L** Mean posterior-anterior (P/A) ratio of OPP incorporation in control and discs expressing *STAT92E-RNAi* in the posterior compartment using *en*-GAL4. Mean and 95% CI, two-tailed Unpaired *t* test, *p*-value < 0.0001 (control: *n* = 8, experiment: *n* = 10). **M, N** OPP incorporation in *egr*-expressing discs, either wild type (**M**) or heterozygous mutant for the *Stat92E*$^{SSC9}$

null allele (**N**). **O** Mean OPP intensity in the pouch of control and the proliferative domain of *egr*-expressing discs (PD$^{egr}$), either wild type or heterozygous mutant for *Stat92E*$^{SSC9}$ null allele. Mean and 95% CI, one-way ANOVA followed by Tukey's post-hoc test for multiple comparisons. (control:*n* = 7, control, *Stat92E*$^{SSC9}$/+: *n* = 7; *egr*:n = 7, *egr, Stat92E*$^{SSC9}$/+: *n* = 7). PD$^{egr}$ vs PD$^{egr}$:*Stat92E*$^{SSC9}$/+: *p*-value = 0.0080 and Control: *Stat92E*$^{SSC9}$/+ vs PD$^{egr}$: *Stat92E*$^{SSC9}$/+: *p*-value = 0.0005. **P, Q** DUAL Control system (DCS) activates *egr*-expression in the *salm*-domain (tracked by pyknotic nuclei and MMP-1 upregulation) and *stat92E-RNAi* in the proliferative domain via *dve*-GAL4 (tracked by UAS-GFP co-expression in **P**), expressing in the pouch and the proximal pouch fold *(Q)*. Protein synthesis visualised by OPP incorporation in control (**P**) and *Stat92E* knockdown (**Q**) disc. **R** Mean OPP intensity in the proliferative domain of control (*DCS*$^{egr}$) and *STAT92E* knockdown (*DCS*$^{egr}$, *Stat92E-RNAi*) wing discs with *egr*-expression in the *salm*-domain. Mean and 95% CI, two-tailed Mann-Whitney's test, *p*-value = 0.0006. (*DCSegr*, control: *n* = 7; *DCSegr, Stat92E-RNAi*: *n* = 8). **S.** Scheme highlighting JAK/STAT positively affecting Pdk1 and S6K. Scale bars: 100 μm. DAPI visualises nuclei. Fluorescence intensities are reported as arbitrary units. Source data in graphs are provided as a Source Data file. Illustrations were created in Biorender Classen, A. (2025) https://BioRender.com/h63vwwi.

with 4% paraformaldehyde for 15 min, rinsed twice in 0.1% PBT, and permeabilized with 0.5% PBT for 15 min. The cuticles were then stained with the Click-iT® cocktail for 30 min at room temperature, protected from light. Further immunohistochemistry analysis and sample mounting was performed as described above.

## EdU labelling
EdU incorporation was performed using the Click-iT Plus EdU Alexa Fluor 647 Imaging Kit. Briefly, larval cuticles were inverted in Schneider's medium and incubated with EdU (10 μM final concentration) at RT for 15 min. Cuticles were then fixed in 4% PFA/PBS for 15 min, washed for 30 min in PBT 0.5%. EdU-Click-iT labelling was performed according to the manufacturer's guidelines. Further immunohistochemistry analysis and sample mounting was performed as described above.

## SA-β-Gal staining
Cell senescence detection kit from Invitrogen (C10850) was used to analyse senescence-associated β-galactosidase activity. Briefly, larval cuticles were inverted in PBS, fixed with 4% PFA, washed with 1% BSA (in PBS) and then incubated in the working solution for 2 h at 37 °C, according to the manufacturer's instructions. Washing steps were performed in PBS and PBS containing 0.1% TritonX-100 (PBT). Further immunohistochemistry analysis and sample mounting was performed as described above.

## Fat body Nile Red staining
Early third instar larvae were collected in PBS and dissected, leaving the gut intact to prevent fat body loss. Inverted cuticles were transferred to an Eppendorf tube and fixed with 4% paraformaldehyde/PBS for 15 min. Samples were washed in 0.1% PBT. Cuticles were incubated with Nile Red (2 μg/mL in PBS) for 1 h, protected from light. Following incubation, samples were washed in PBS, the fat body was dissected and mounted as described above.

## Hemolymph sample preparation
Fifteen larvae were collected and thoroughly washed with Milli-Q water to remove any fly food particles. Care was taken to ensure no food particles remained on the larvae surfaces, and they were dried using Kim Tech paper wipes. Each larva was punctured in the centre using forceps and transferred to a 0.5 mL microcentrifuge tube with three 1 mm holes at the bottom. The 0.5 mL microcentrifuge tube was then placed into a pre-cooled 1.5 mL microcentrifuge tube. One larva at a time was processed in this assembly and centrifuged in a microfuge for 10 seconds. After each centrifugation, the larval carcass was removed

from the 0.5 mL microcentrifuge tube to prevent blockage of the holes. Hemolymph isolated from the 15 larvae was collected in the bottom 1.5 mL tube. From the total collected hemolymph, 8 μL was transferred to a fresh 1.5 mL tube, and 10 μL of ultrapure Milli-Q water, 30 μL of methanol, an internal standard, and 50 μL of MTBE were added. The solution was mixed thoroughly and centrifuged at 1000 g for 10 minutes at 4 °C. After centrifugation, the organic and polar phases were collected separately in different tubes for metabolite measurement.

## Hemolymph metabolomic analyses
Non-targeted analysis of polar metabolites by LC-MS was carried out as described previously[105] using an Agilent 1290 Infinity II UHPLC in line with a Bruker Impact II QTOF-MS operating in negative and positive ion mode. Scan range was from 20 to 1050 Da, and mass calibration was performed at the beginning of each run. LC separation was on a Waters Atlantis Premier BEH ZHILIC column (100 ×2.1 mm, 1.7 μm particles), buffer A was 20 mM ammonium carbonate and 5 μM medronic acid in milliQ H2O, and buffer B was 90:10 acetonitrile:buffer A and the solvent gradient was from 95% to 55% buffer B over 14 min. Flow rate was 180 μL/min, column temperature was 35 °C, autosampler temperature was 5 °C and injection volume was 3 μL. Data processing, including feature detection, feature deconvolution, and annotation of features, was performed using MetaboScape (version 2023b). Quantile normalisation was performed to minimise sample size effects, and further statistical processing were performed with R. Metabolites with missing values were eliminated from the dataset. Both the Shapiro-Wilk test and the Q-Q plot, showed that the data was not normally distributed. Therefore, a two-sided, unpaired Wilcoxon rank-sum test was performed on the quantile-normalised data. Only metabolites with at least < 0.75 and > 1.5-fold change and a *p*-value < 0.19 and were selected for further analysis and were displayed in a heatmap with row-wise normalization (Fig. 3L).

## Thin layer chromatography (TLC)
Fat bodies were dissected in cold PBS from third instar (L3) control (rn-GAL4, tubGAL80ts) and *egr*-expressing (rn-GAL4, UAS-egr, tub-GAL80ts)larvae, and from wild type larvae starved for 24 h. Three fat bodies were pooled per sample and immediately transferred into 100 μL chloroform: methanol (3:1) solution, followed by storage on ice. Samples were mechanically homogenised and centrifuged at 15,000 × *g* for 5 min at 4 °C. For semi-quantitative analysis of triglycerides, a standard curve was prepared using lard as a reference material, dissolved in chloroform: methanol (3:1) at the following concentrations: STD1 (75 μg/μL); STD2 (60 μg/μL); STD3 (50 μg/μL);

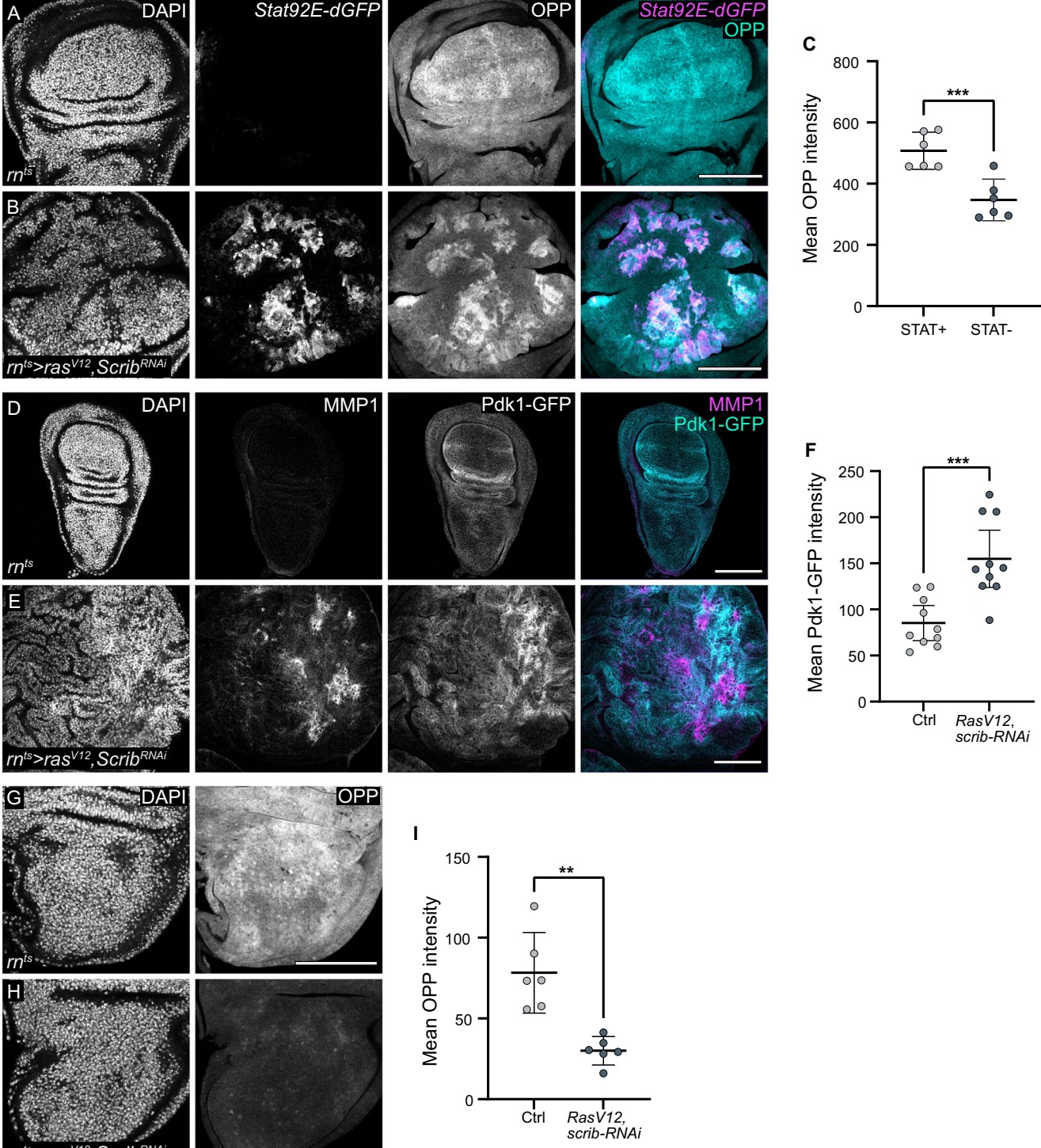

**Fig. 7 | Pdk1 upregulation, protein translation and JAK/STAT signalling are linked in tumour growth. A**, **B** Control (**A**) and wing disc expressing *Ras^{V12}, scrib-RNAi* (**B**) for 44 h starting at Day6 AED. OPP incorporation (cyan/grey) and JAK/STAT-reporter *10xStat92E-dGFP* (magenta/grey). **C** Mean OPP intensity in the *10xStat92E-dGFP* positive and negative region within a *Ras^{V12}, scrib-RNAi*-expressing wing discs. Mean and 95% CI, two-tailed Paired *t* test, *p*-value = 0.0006 (control: *n* = 6, experiment: *n* = 6). **D**, **E** Control and wing disc expressing *Ras^{V12}, scrib-RNAi*. Pdk1-GFP (cyan/grey) in control (**D**) and *Ras^{V12}, scrib-RNAi*-expressing wing disc (**E**), along with MMP1 staining to visualises JNK-activity (magenta/grey). **F** Mean Pdk1- GFP intensity in the control pouch and *Ras^{V12}, scrib-RNAi*-expressing pouch. Mean and 95% CI, two-tailed Unaired *t* test, *p*-value = 0.0004 (control: *n* = 10, experiment: *n* = 10). **G**, **H** OPP incorporation in the notum of control (**G**) and wing disc expressing *Ras^{V12}, scrib-RNAi* (**H**). **I** Mean OPP intensity in the notum of control and wing disc expressing *Ras^{V12}, scrib-RNAi*. Mean and 95% CI, two-tailed Welch's *t* test, *p*-value = 0.0031 (control: *n* = 6, experiment: *n* = 6). Scale bars: 100 μm. DAPI visualises nuclei. Fluorescence intensities are reported as arbitrary units. Source data in graphs are provided as a Source Data file.

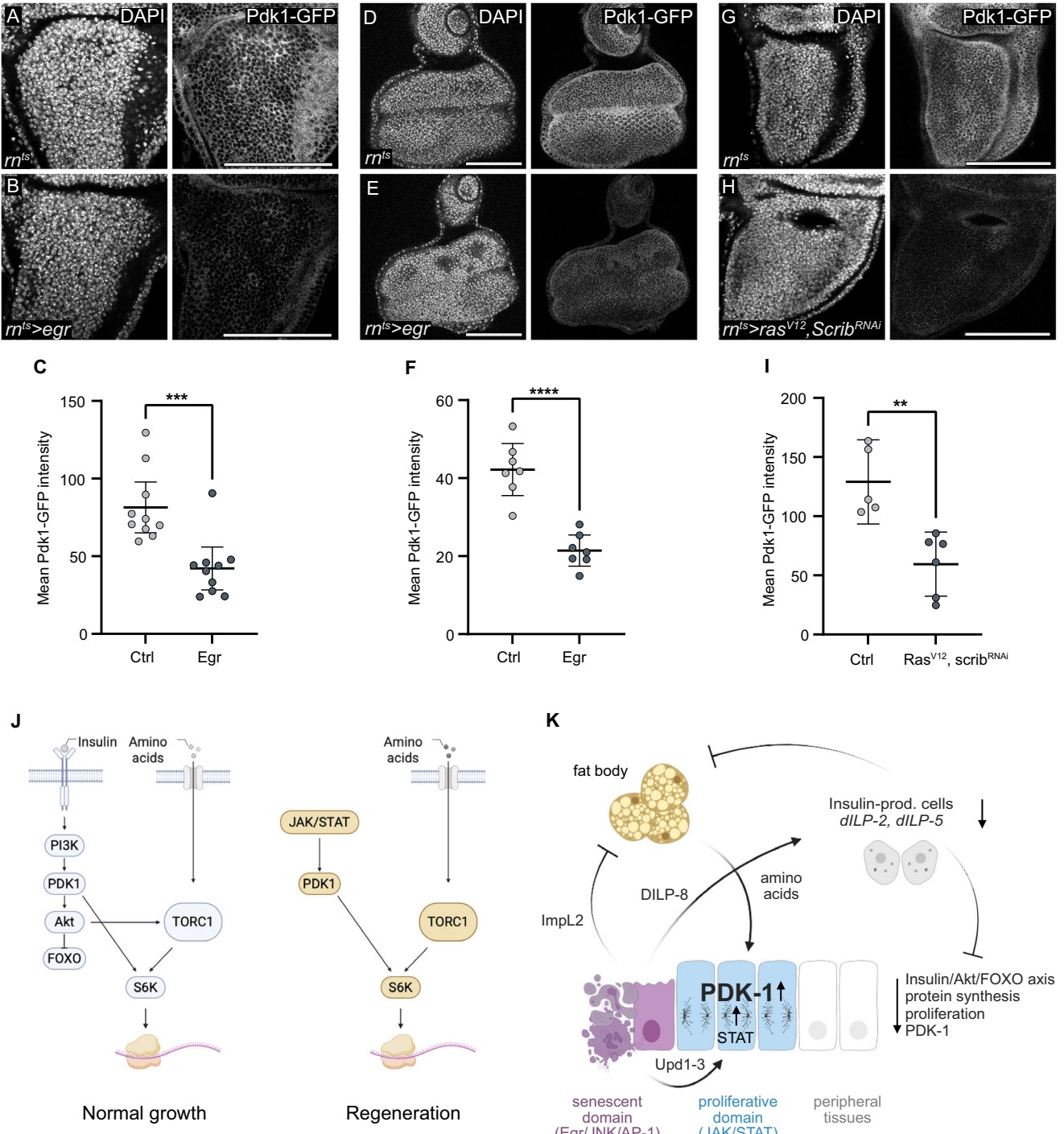

**Fig. 8 | Pdk1 downregulation in peripheral discs correlates with systemic growth restriction. A, B** Pdk1-GFP in the notum of control (**A**) and *egr*-expressing discs (**B**). **C** Mean Pdk1-GFP intensity in the notum of control (**A**) and *egr*-expressing discs (**B**). Mean and 95% CI, two-tailed Mann-Whitney's test, *p*-value = 0.0007 (control: *n* = 10, experiment: *n* = 10). **D, E** Pdk1-GFP in eye imaginal disc, dissected from larvae with control (**D**) or *egr*-expressing (**E**) wing discs. **F** Mean Pdk1-GFP intensity in eye discs, dissected from larvae with control (**D**) or *egr*-expressing (**E**) wing discs. Mean and 95% CI, two-tailed Unpaired *t* test, *p*-value < 0.0001 (control: *n* = 7, experiment: *n* = 7). **G, H** Pdk1-GFP in the notum of control (**G**) and wing disc expressing *Ras^V12, scrib-RNAi* (**H**). **I** Mean Pdk1-GFP intensity in the notum of control and wing disc expressing *Ras^V12, scrib-RNAi*. Mean and 95% CI, two-tailed Unpaired *t* test, *p*-value = 0.0022 (control: *n* = 5, experiment: *n* = 6). **J** Model summary illustrating growth mediated by S6K activation via canonical Insulin signalling (Insulin/AKT/FOXO and mTORC1) during normal developmental (left). Growth in the

proliferative domain is maintained in an insulin-restricted environment through the JAK/STAT-Pdk1-S6K axis, supported by mTORC1 signalling. **K** Model summary of local and systemic metabolic adaptations supporting growth in the proliferative domain. Inter-organ signalling is initiated by senescent-like cells in the high JNK signalling domain at the centre of tissue damage via secreted Dilp8, ImpL2 and Upd1,2,3. Dilp8 reduces insulin-like peptide expression by acting on IPCs, thereby restricting systemic insulin signalling in peripheral tissues. ImpL2 acts on fat body, and combined both mechanisms facilitates nutrient mobilisation from stores. Secreted Unpaireds activate JAK/STAT in the nearby proliferative domain, upre-gulating Pdk1 and instructing S6K activation. Levels of Pdk1 in the proliferative and peripheral regions determine tissue growth in an insulin-restricted environment. Scale bars: 100 μm. DAPI visualises nuclei. Fluorescence intensities are reported as arbitrary units. Source data in graphs are provided as a Source Data file. Illustrations were created in Biorender Classen, A. (2025) https://BioRender.com/h63vwwi.

STD4 (37.5 μg/μL); STD5 (25 μg/μL); STD6 (15 μg/μL); STD7 (10 μg/μL). A blank control containing only the solvent mixture (chloroform:methanol (3:1)) was included. Samples and standards were loaded onto a glass silica gel TLC plate (Millipore). The mobile phase consisted of a hexane:diethyl ether (4:1) solvent system. Chromatography was performed, and after air-drying, the TLC plate was treated with ceric ammonium heptamolybdate (CAM) staining solution and developed at 80 °C, with band development monitored at 5 min intervals. Images were captured using a gel documentation system (gelONE); image analysis and quantification were performed using ImageJ. Automatic image thresholding was applied using the Otsu method, and regions of interest (ROIs) were defined for each band. Integrated intensity analysis was performed, and triglyceride density was quantified based on the standard curve.

### Real-time PCR

Third instar (L3) control (rn-GAL4, tubGAL80ts), egr expressing (rn-GAL4, UAS-egr, tubGAL80ts), and wild type larvae starved for 24 h larvae were carefully collected from the food and pooled in groups of three male larvae per sample. Samples were homogenised in 100 μL of TRIzol, and RNA was extracted using chloroform (20 μL), followed by precipitation with 50 μL of isopropanol. The purified RNA was suspended in nuclease-free water, subjected to DNase treatment and reverse transcription was performed using RevertAid Reverse Transcriptase (Thermo Fisher Scientific) according to the manufacturer's guidelines. qPCR was performed using the Blue S'Green qPCR Kit Separate ROX (Biozym) on a LightCycler 480 (Roche). Standard dilution series (1:5) were generated for each primer pair to determine primer efficiency based on the regression curve. The following primers were used (for, rev 5′–3′): Rpl1 (TCCACCTTGAAGAAGGGCTA, TTGCGGATCTCCTCAGACTT), Ilp2 (ATCCCGTGATTCCACCACAAG, GCGGTTCCGATATCGAGTTA), Ilp5 (GCCTTGATGGACATGCTGA, AGCTATCCAAATCCGCCA)[106,107]. Gene expression levels were normalised to the reference gene Rpl1, and relative expression was calculated using the Pfaffl method. Statistical outliers were identified using the ROUT test (Q = 1%) and were removed to maintain data consistency. Statistical analysis was performed using the Kruskal-Wallis test.

### Western blot analysis

Fat bodies were dissected in cold PBS from third-instar (L3) control (rn-GAL4, tubGAL80ts) and egr-expressing (rn-GAL4, UAS-egr, tubGAL80ts) larvae, and from wild-type larvae starved for 24 h. Five fat bodies were lysed on ice (300 mM NaCl, 50 mM Tris-HCl ph 7,5, 1% Triton X-100, 0.1 mM EDTA, 0.1% SDS, 5% Glycerol, Roche Complete protease inhibitor cocktail and Complete phosphate inhibitor cocktail). Tissues were further homogenised and centrifuged at $6000 \times g$ for 3 min at 4 °C. Supernatants were incubated in Laemmli buffer at 85 °C for 5 min. Samples were loaded onto Mini-PROTEAN TGX gels (4–15%) corresponding to one fat body per lane. After electrophoresis, proteins were transferred onto a nitrocellulose membrane, blocked in 1 × Tris-buffered saline containing bovine serum albumin, incubated overnight with primary antibodies at 4 °C and secondary antibody for 1 h at RT. Proteins were visualised using SuperSignal West Femto Maximum Sensitivity Substrate and a Bio-Rad ChemiDoc-MP imaging system.

### Image quantification and statistical analysis - General comments

For all quantifications, control and experimental samples were processed together and imaged in parallel, using the same confocal settings. Images were processed, analysed and quantified using tools in Fiji (ImageJ 2.14)[108]. Care was taken to apply consistent methods to control and experimental samples (i.e., number of projected sections, thresholding methods, processing) for image analysis and quantifications. See Supplementary Fig. S1 for macros used in FIJI during image segmentation and quantification. Further details on segmentation and

quantification are provided below. Figure panels were assembled using Affinity Designer 2.4.0. Statistical analyses were performed in Graphpad Prism. Illustrations were created in Biorender Classen, A. (2025) https://BioRender.com/h63vwwi. To quantify fluorescence intensities inside the proliferative domain (PD$^{egr}$), a mask of the high JNK-signalling domain region was generated, followed by creating a 20 μm band on the outside to mark the proliferative domain. Fluorescence intensities in proliferative domains was always compared to fluorescence intensities within the pouch domain of control discs, unless otherwise noted.

### Quantification of cycling cells using EdU incorporation

(a) **EdU quantification in notum:** An xy-section with the maximum number of notum epithelial cells was selected, excluding any myoblast cells. A mask was created from the DAPI staining to represent nuclei and saved as an ROI. A binary operation (AND) was used to compute a notum DAPI mask with the aid of a manually selected notum region (FileS1, Macro-1L(a)). This mask was then used to measure EdU intensity.

(b) **EdU quantification within 10μm bands:** In egr-expressing discs, TRE-RFP expression was used to generate a mask of the high JNK-signalling domain (FileS1, Macro-1D(a)). The boundary of this mask defined the regions inside and outside the high JNK region. Five 10μm bands were created outside the high JNK-signalling domain, and three 10μm bands inside of the JNK-signalling domain. Also applying a DAPI mask to each band (FileS1, Macro-1D(b)), the mean EdU intensity in each of the 10μm bands was measured. Consistent with previous studies[24,26], the highest EdU intensities were observed in 3 bands outside the JNK-signalling domain, which was denoted as the proliferative domain.

(c) **EdU quantification in eye disc:** For the eye disc, an EdU mask (FileS1, Macro-2T(a)) was generated using the EdU staining, which was then applied to measure EdU intensity in the entire visible eye disc.

### Quantification of Upd3-LacZ levels

A maximum projection of TRE-RFP expression was used to generate a mask of the high JNK-signalling domain in egr-expressing disc (FileS1, Macro-S3H(a)). Upd3-LacZ intensity was measured within the TRE-RFP mask of the high JNK-signalling domain in egr-expressing disc, while for the control disc, Upd3-LacZ intensity was measured in the pouch.

### Quantification of ImpL2-GFP levels

A pouch mask was generated using the maximum projection of an anti-Nubbin staining (FileS1, Macro-3H(a)) for both control and egr-expressing discs. ImpL2-GFP intensities were measured within the Nubbin mask on the sum projection of the ImpL2-GFP signals within the stack (seven slices in each disc).

### Quantification of OPP levels

(a) **in different regions of wing imaginal disc:** In the control disc, mean OPP intensity was measured separately in the pouch, hinge, and notum regions. In egr-expressing discs, mean OPP intensity was measured in the high JNK signalling region (TRE-RFP mask), the proliferative region (20 μm band around the TRE-RFP mask), and the notum region. In control discs, the pouch, hinge, and notum regions were selected using wing fold landmarks. In egr-expressing discs, a max projection of TRE-RFP intensities was used to create a mask of the high JNK signalling domain (FileS1, Macro-1G(a)), which was applied to measure OPP intensity in the high JNK region. A 20μm band outside the high JNK region was generated to mark the

proliferative region, and OPP intensity in this domain was measured. For the notum, a manual selection along the first notum fold and outlining the edges of the notum was carried out.

(b) **in the eye disc:** For the eye-antennal disc, the DAPI staining was used to generate a mask of the tissue outline (FileS1, Macro-2Q(a)) and combined with a manual selection of the eye disc to generate a region of interest (ROI) within which we measured OPP intensity.

## Quantification of nuclear dFOXO levels

In control discs, a slice with the maximum number of pouch cells was selected, and a DAPI mask (Supplementary File S1, Macro-4G(a)) was generated and combined with manual pouch selection to create a pouch nuclear mask. This mask was used to measure mean dFOXO intensity in the pouch nucleus. In *egr*-expressing discs, a single slice from the Z-stack with the maximum number of cells in the proliferative domain was selected. A max projection of TRE-RFP intensities was used to create a mask of the high JNK signalling domain, and a 20 μm band was created to locate the proliferative region. A DAPI mask (Supplementary File S1, Macro-4G(b)) was generated from the selected slice and combined with the 20 μm band to create a new mask for the proliferative cell nuclei, which was used to measure dFOXO intensity in the nucleus. For the analysis of nuclear dFOXO in the notum, a Z-slice containing the maximum number of notum cells was selected. A DAPI mask (Supplementary File S1, Macro-2F(a)) was generated and combined with manual selection of notum, guided by wing fold, to create a notum nuclear mask. Please note that discs and fat body were treated with Leptomycin B (413 nM) for 30 min before dissection to block nuclear export.

## Quantification of Pdk1 GFP, p-S6 and OPP levels in the en-GAL4 domain

The slice with the maximum number of pouch cells was selected, and masks of the anterior and posterior pouch compartments were generated by segmenting the enGAL4, UAS-RFP expression domain. Levels of p-S6(Supplementary File S1, Macro 6 F(a)) and Pdk1-GFP intensities were measured within these masks, and the P/A ratio (en-GAL4 region/non-en-GAL4 region) was calculated. Mean OPP intensity in the control and experimental pouch en-GAL4 domain was measured and reported.

## Quantification of 10xStat92E-GFP levels in the fat body

A single slice from the Z-stack with the maximum number of nuclei was selected. A 200 × 200 pixel square region was defined and used as a standard area for measurement of GFP intensities.

## Quantification of lipid droplet size and shape

A single slice from the Z-stack through the most anterior fat body containing the maximum number of nuclei was selected. A published STAR protocol was applied to measure lipid droplet area and the 'roundness' of lipid droplets (defined as the relationship between area and length of major droplet axis) in both control and experimental samples[109].

## Reporting summary

Further information on research design is available in the Nature Portfolio Reporting Summary linked to this article.

## Data availability

All data, workflows and FIJI algorithms necessary to interpret the data are included within the manuscript. Because of the large data set size, microscopy images and segmentation results are available upon request. Requests should be addressed to and will be fulfilled within a week by AKC. Source data are provided in this paper.

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

## Acknowledgements

We thank the staff of the Life Imaging Centre (LIC) in the Hilde Mangold House (HMH) of the Albert-Ludwigs-University of Freiburg for help with their confocal microscopy resources, and the excellent support in image recording. We specifically thank the DFG for supporting our imaging work through project number 414136422. We thank the Lighthouse Core Facility and its staff, as well as Elisabeth Savkova, Janhvi Jaiswal and Carlo Crucianelli for help with our experimental work. We thank Robin Harris, Susumi Hirobayashi, Hugo Stocker, Aurelio Teleman and Hong Xu, Iswar Hariharan, Dirk Bohmann for sharing reagents and the Bloomington *Drosophila* Stock Centre (BDSC), the Vienna *Drosophila* Stock Collection (VDRC), the University of Zurich ORFeome Project (FlyORF), the Developmental Studies Hybridoma Bank (DSHB) and the Monoclonal Antibody Core Facility at the Helmholtz Zentrum Munich for providing fly stocks and antibodies. We thank the IMPRS-EBM and SGBM graduate schools for supporting our doctoral researchers. Funding for this work

was provided by the Deutsche Forschungsgemeinschaft (DFG, German Research Foundation) under Germany's Excellence Strategy (CIBSS – EXC-2189) to K.K. and A.K.C., and by the DFG Heisenberg Programme to A.K.C. (668189) and to K.K. (544402801), as well as DFG grants to AKC (667603) and to K.K. (529943809). Funding was furthermore provided by the Boehringer Ingelheim Foundation (BIF Plus3 & Rise Up) to A.K.C. The funders had no role in study design, data collection and analysis, decision to publish, or preparation of the manuscript.

## Author contributions

Conceptualisation A.V.M. and A.K.C.; Data curation A.V.M., L.N.e, L.N.o, L.H., and A.K.C.; Formal analysis A.V.M., L.N.e, L.H., J.B., and A.K.C.; Funding acquisition K.K. and A.K.C.; Investigation A.V.M., L.N.e, L.N.o, A.S., and A.K.C.; Methodology A.V.M., L.N.e, L.N.o, A.S., I.G., J.B., and A.K.C.; Project administration I.G. and A.K.C.; Supervision K.K. and A.K.C.; Validation A.V.M., L.N.e, L.N.o, A.S., K.K., and A.K.C.; Visualisation A.V.M., L.N.e, and L.H.; Writing – original draft A.V.M. and A.K.C.; Writing – review & editing A.V.M., K.K., and A.K.C.

## Funding

## Competing interests

The authors declare no competing interests.
