## [Peer Review file · Nature Communications]

A JAK/STAT-Pdk1-S6K axis bypasses systemic growth restrictions to promote regeneration

Corresponding Author: Professor Anne-Kathrin Classen

Version 0:

Reviewer comments:

Reviewer #1

(Remarks to the Author)

I view the manuscript as a loosely connected collection of potentially related data. The authors propose a mechanism based on correlative data without providing a clear demonstration of causality. Moreover, even if the data supports their claims, the conceptual advance presented in this manuscript appears incremental.

Major Issues:

1. A fundamental problem with this manuscript is the lack of genetic manipulation (except for the PDK1 heterozygous condition, which presents its own issues, as discussed later) alongside Eiger expression driven by rotund-GAL4, tub-GAL80ts. Since the development of the ablation system by the Hariharan lab at UC-Berkeley, independent manipulation of genes and ablation has been a challenge. However, the Harris lab at Arizona State University developed a system enabling simultaneous cell ablation and gene manipulation (Harris et al., eLife 2020). Without employing such a genetic approach, the authors failed to demonstrate a causal relationship among genes/proteins. Their claims of causality throughout the text are overstated.
2. Except for the PDK1 results in Figs. 5-6, much of the data lacks novelty. The authors should more clearly distinguish their contributions from previously published work, particularly that of the Hariharan lab, which has reported many similar findings.
3. There is a logical inconsistency regarding the relationship between OPP, EdU, p-S6, and PDK1. In Figure 1, there is no increase in OPP signals in proliferating cells, yet in later figures, the authors suggest that PDK1 enhances OPP signals. If PDK1 genuinely functions in regenerating cells, OPP should be upregulated in these cells.
4. The authors mention that Dilp8 or Eiger regulates insulin signaling, but then conclude that ImpL2 is responsible for insulin reduction. How do they exclude the roles of Dilp8 and Eiger?
5. In Figure 3, the relevance of much of the data is unclear. mTOR may be permissive rather than instructive in this context. For instance, even if they inhibit actin or tubulin, they might observe similar results. Moreover, the significant effect of rapamycin on the control condition is noteworthy. Additionally, the authors mention urea production, but they should be aware that the urea cycle in flies is defective due to the lack of ornithine transcarbamylase. This makes their claim misleading.
6. It is unclear why the authors are showing STAT activation in the fat body in Figure 4. Additionally, they assess lipid droplets in the fat body, but they should measure TAG levels for a more accurate assessment.
7. In Figure 5, much of the data again appears to be merely correlative. Does STAT induce p-S6 through PDK1?
8. In Figure 6, the authors use PDK1 heterozygosity, which may affect JNK signaling in the ablated region, as PDK1 is known to regulate JNK. PDK1 heterozygosity could also influence other processes in distant organs, such as the fat body. As mentioned earlier, a tissue-specific approach is necessary.
9. The authors claim that other organs exhibit lower PDK1 levels, but in Figure 4, they demonstrate STAT activation in the fat body, which is mechanistically confusing.

10. The high levels of PIP3, which is induced upon insulin binding to its receptor, in regenerating tissues suggest that the effects of FOXO are not due to ImpL2, since ImpL2 inhibits insulin's binding to its receptor. This contradicts their claim. If they think PI3K is activated not through insulin, they need to demonstrate how it occurs.

11. If their claim is correct, the novel aspect of their findings is the role of JAK-STAT and PDK1 in regenerating cells. It is therefore crucial to clarify how JAK-STAT regulates PDK1. Without this mechanistic insight, the contribution of this paper remains shallow.

12. It would be ideal to demonstrate the spatial dynamics of ImpL2 and UPD3, specifically whether they reach other tissues, such as the fat body.

Specific Issues:

13. Some figures, such as those showing FOXO and PIP3 imaging, require higher magnification for better clarity.

14. The rotund-GAL4 driver promotes expression in the myoblasts beneath the notum. Some observations described in Figure 1 could be attributed to gene expression in myoblasts.

15. In the text, they incorrectly refer to beta-gal expression, which should be beta-gal activity.

Optional Point to Address:

In the introduction and discussion, the authors frame their findings in the context of tumorigenesis and tissue damage. Despite this, throughout the manuscript, they rely solely on Eiger expressed via rotund-GAL4, which limits the scope of their conclusions. While not essential to support their claims, it would be interesting to explore whether PDK1 functions in other contexts, such as rpr-mediated ablation and tumorigenesis.

Reviewer #2

(Remarks to the Author)

In this manuscript the authors address the interesting observation that when imaginal discs are damaged, proliferation occurs right next to the wound to regenerate missing tissue but is suppressed in other parts of the wing disc. From their experiments they propose that proliferation is generally arrested because of reduced insulin signaling. However, in the proliferating region, JAK/STAT signaling activates Pdk1. Additionally the catabolism of the fat body results in increased circulating amino acids which promotes Tor activation in the regenerating tissue. Together, Pdk1 and Torc1 promote cell proliferation in this region.

This study addresses an important and interesting question - how proliferation is activated in one part of the tissue and suppressed in another. The authors make several interesting observations. However, there are issues of data presentation and data interpretation in the study as currently presented. Also, as discussed at the end of this review, the model proposed by the authors is not substantiated by the data presented – alternative interpretations are possible.

Detailed comments

In Figure 1, the authors document the phenomenon nicely of proliferation in the eiger expressing domain and a suppression of proliferation in the rest of the disc. The authors present this as a completely novel discovery. However, the suppression of proliferation in non-regenerating portions of the disc was first shown by Sustar and Schubiger (2005) in transplantation experiments. Even for the eiger ablation system that the authors are using, the suppression of proliferation in non-regenerating parts of the disc was shown in multiple ways in the original paper that described the eiger ablation system (Smith-Bolton et al 2009).

Figure 2 addresses the effects of tissue ablation in the wing disc on other remote tissues. There are however, many issues of data presentation and interpretation.

In panels A and B the authors seem to think (line 697) that “the yellow arrowhead points to the expected location of the mitotic wave just anterior to the morphogenetic furrow”. This is incorrect. The arrowhead points to what is referred to historically as the “second mitotic wave” (SMW) which is a row of synchronous S-phases immediately posterior to the morphogenetic furrow. That only a portion of this row can be seen in panel A (it typically runs across the disc) indicates either that the preparation is not flat or that the EdU incorporation has not occurred through the entire preparation. The authors should familiarize themselves with the pattern of EdU incorporation in the normal eye disc. It would seem that the SMW is gone in panel B (if so, this is interesting), but this needs to be shown properly. Quantifying EdU incorporation in the yellow box does not really achieve anything clear because it includes the G1-arrested region in the furrow and the EdU incorporation anterior to the furrow (top right hand corner of the box).

The eye disc also has different portions which are actively proliferating, paused in G1 and postmitotic. It is unclear which region of the eye disc is being measured in the OPP quantification in panel F. Each region in the wild-type should probably be compared to the corresponding region in the mutant condition.

The leg disc images seem more convincing– not sure why these data have not been quantified and have been relegated to a supplementary figure. One issue with this experiment is that *rn-Gal4* is also expressed in the leg disc. The authors should check whether tissue is being killed in the leg disc. If so, this cannot be thought of as a response in tissue that is distant from

the site of damage.

In panels I, J and K, the authors claim that the levels of nuclear FOXO (as assessed by FOXO-GFP) in the fat body are increased when there is tissue ablation in the wing disc. Their FOXO-GFP is primarily localized in the nucleus. There are many published reports of FOXO staining (using an antibody) in the fat body in non-starved L3 larvae and FOXO is present throughout the cell (for example Figure 2 in Koyama et al. 2014). Thus, the FOXO-GFP reagent does not seem to accurately reflect FOXO localization and it is therefore questionable if these small differences in the intensity really reflect what is happening to FOXO itself.

In Figure 3, the images showing the increase in P-S6 are convincing. The authors previous work suggests that the cells in the proliferating zone are proliferating faster. If this is the case, then the additional sensitivity to rapamycin might simply be because these cells are proliferating faster. Is the proliferating region also sensitive to inhibiting other growth pathways? Without testing that, it is hard to claim that the TOR pathway is especially important for these cells.

From the data in Panel L the authors conclude (line 253) that “Our results revealed an enrichment of many amino acids in egr-expressing hemolymph.” But their data also show amino acids and peptides that are DECREASED in egr-expressing hemolymph (Asp Lys, Asn Gln Lys, Glu Ile Arg). Thus the data presented do not seem consistent with a general increase in amino acid levels in the hemolymph when *eiger* is expressed in the wing disc.

In Figure 4 The authors show nicely that there is increased JAK/STAT signaling in the fat body.

Two things I find confusing about the data shown in this experiment:

- 1) In panels A and B, both the size and roundness of lipid droplets increases in the *eiger* condition. What do the authors mean by roundness? Please explain.
- 2) If the lipid droplets are larger in the *eiger* condition, how is this consistent with their hypothesis that nutrients are being mobilized from the fat body to fuel the regeneration. Would this not lead to a depletion of triglycerides in the fat body? Please explain.

In Figure 5 the authors claim that “both cytoplasmic and nuclear levels of FOXO were elevated”. How can we see this in the low magnification images shown? Also note the earlier point of how FOXO-GFP may not accurately reflect FOXO localization.

The data showing Pdk1-GFP is convincing. (It would be nice if the authors stated clearly if the GFP is a transcriptional reporter or a protein fusion in each case that they use these reagents).

The data shown in Figure 6 are convincing – that Pdk1 promotes growth and cell cycle progression, and that regenerative proliferation is sensitive to Pdk1 levels.

In summary, the authors do show that the JAK/STAT pathway and Pdk1 are important for the proliferation of cells around the *eiger*-damaged tissue. Since they have not looked at other pathways (e.g. RTK, Myc, Hippo) I don't see how they can conclude that this pathway is especially important – similar results may be observed if other pathways are tested.

Also if the JAK/STAT ligands can reach distant sites such as the fat body, why is this pathway not getting activated in other parts of the wing disc such as the notum which is much closer to the site of release. Why is this pathway only operating in the proliferating zone?

The data presented also have many technical issues – especially Figures 3 -5. These issues need to be sorted out.

The data presented in the UMAPs in supplemental figure S3 need to be explained better. From the figure legend it seems that the points represent cells from both control and *eiger*-expressing discs. Without showing the two conditions separately, how do we know whether these expressing cells are from control or *eiger*-expressing discs? Also *ImpL2* seems to be expressed most highly in a part of the disc that is very different from where all the wound ligands are expressed. What is happening with *ImpL2*?

Another point is that the referencing of the literature needs to be improved. The authors cite their own work a lot but do not cite previous work that has made the same observations. This is observed in multiple places in the manuscript. For example (line 191) “We previously reported that *eiger*-expressing larvae also exhibit a significant delay in initiating pupariation (La Fortezza et al 2016)”. This observation has a long history. The delay following tissue damage from radiation was first described by Hussey (1927) and from damage to imaginal discs by Simpson et al (1980). Even the first description of the pupariation delay after damaging imaginal discs by *eiger* expression was first described by Smith-Bolton et al. (2009) - long before La Fortezza et al.

Reviewer #3

(Remarks to the Author)

This is an interesting study that looks at how metabolism of proximal and distal tissues is altered when a wound is inflicted in a tissue. Interestingly, it shows that the regions proximal to the wound are able to continue proliferating and growing due to up-regulation of PDK1 expression, despite a likely systemic drop in insulin signaling. This study will be interesting for a broad audience studying growth, insulin/TOR signaling, tissue damage and regeneration, and translation. Overall the data

are of good quality. I only have minor issues as suggestions for improving the manuscript.

Minor Issues

1. Suppl. Fig. 1D-F: It would be helpful to quantify the proportion of cells that are in the different phases of the cell cycle in the control pouch and in the $m>egr$ region
2. Line 146 says "They induce compensatory proliferation in surrounding cells, which can be detected by rapid EdU incorporation (Fig 1A,B)"
To me the EdU does not look higher around the wound in panel B compared to the control disc in panel A ? Either quantify or rephrase the sentence?
Or is it rather that proliferation is inhibited in the rest of the $m>egr$ disc ?
3. It would be good to solidify that there's reduced systemic insulin signaling in animals with wing lesions compared to controls. E.g. do a WB on pAkt on salivary glands or fat body or whole larvae?
4. Fig 3I-K: I see clearly the upregulation of CG5535 in the senescent domain, but it's not so visually obvious in the proliferative domain. How is the senescent domain marked, to make sure it's not getting included in the quantification of the proliferative domain? Could it be that the mild increase in CG5535-GFP shown in the quantification in panel K is actually coming from the senescent domain? Same for Fig S3E-F.
5. Fig 4A-D, the authors observe some morphological changes to lipid droplets in the fat body. They appear to actually get larger and stain more intensely with Nile red. They then conclude "the observed hemolymph changes may be derived from fat body catabolism". I would rather guess these changes look like lipid anabolism, rather than catabolism? But the "may" is critical because I suppose it is possible. But the 'may' then disappears from the rest of the manuscript, as this conclusion gets taken for granted. For instance, at the end of the next paragraph: "These observations indicate that the production of ImpL2 and Upd cytokines from egr -expressing discs are likely responsible to drive nutrient mobilization from the fat body..." If the conclusion that the fat body mobilizes nutrients is important, it should be shown solidly. Otherwise, the respective sentences should be rephrased, also in the discussion.
6. Line 308: "If egr -expressing discs induce low systemic Insulin signaling..."
It would be good to rephrase this sentence because "induce" usually means to increase, so it sounds like this sentence is saying that insulin signaling is going up rather than down. Perhaps "cause" instead of "induce"?
7. One major conclusion is that the proliferative zone is able to continue proliferating because JAK/STAT signaling induces PDK1 expression. Can this be shown epistatically? ie if JAK/STAT signaling is inhibited (e.g. with loss-of-function mitotic clones) in the proliferative area, does this block PDK1 upregulation? (I don't know if this experiment is logistically possible.)

Version 1:

Reviewer comments:

Reviewer #1

(Remarks to the Author)

The authors have addressed some of my previous comments; however, they have failed to address the most fundamental issues raised in my original review.

1. In my initial review, I pointed out that one of the core problems was that PDK involvement was shown only in a correlative manner. In response, the authors introduced the Dual Control approach. Unfortunately, the data presented are not convincing. Figures 5L–M and 6P–Q are particularly problematic and lack clarity. Notably, even regions outside of the wing pouch exhibit reduced signal, making it difficult to interpret the results. The authors need to clarify which region is ablated and which region expresses RNAi. Without such demarcation, the conclusions drawn are unreliable.

11. The authors did not address the key mechanistic question of how Stat induces PDK. This remains the most critical unresolved issue in the current manuscript.

12. I had requested clarification of the spatial dynamics of Upd3, which is crucial to their hypothesis. The authors speculate that Upd ligands activate PDK in proliferating tissues such as the wing disc and also reach peripheral tissues like the fat body. However, they also claim that PDK is suppressed in these peripheral tissues. This is logically inconsistent. The dynamics of Upd ligands (comment 12) and their relationship with PDK activation (comment 11) are essential for supporting the central claim of this study. Without addressing these points, the manuscript offers only marginal novelty.

13. I requested higher-magnification images of FOXO-GFP (Fig. 2D–E, etc.) and PIP3, but the authors did not provide them. Higher magnification is necessary for readers to properly assess the data. The current images are insufficient; figures should be interpretable without relying solely on quantitative plots.
Lastly, several points made by the authors in the rebuttal should be explicitly reflected in the revised manuscript text.

Reviewer #2

(Remarks to the Author)

The revised manuscript is much improved. The authors have addressed most of my concerns. This now makes a nice story that is supported adequately by the data. The data are well presented. I still have some issues that can be addressed mostly by textual changes. These are:

- 1) The authors should clarify the nature of the PDK1-GFP reagent. I assume this is construct that has a GFP fused to the endogenous PDK1. Is this correct? Or is it simply a transcriptional reporter? It is important to explain this clearly. Otherwise it is hard to know whether it reflects PDK1 transcription, PDK1 protein levels, or both. I was not able to find this explanation anywhere.
- 2) I think the conclusion that PDK1 is sufficient for regenerative proliferation (line 283) is overstated. The wound region makes many factors that likely function together with PDK1 to promote regenerative growth. Sufficiency implies that it can do it on its own.
- 3) The authors conclude that the increase in some amino acids in the hemolymph is a result of their mobilization from the fat body. This is likely but other explanations are formally possible. The authors show lipolysis in the fat body and an increase in amino acids in the hemolymph. The amino acids could be coming from somewhere else. Also, I could not see an increase in free fatty acids in the hemolymph. Is this because the metabolomics only identifies water-soluble molecules? Since there is an inference made from the metabolomics that the changes in the hemolymph result from the breakdown of material in the fat body, why is there no evidence of increased levels of free fatty acids in the hemolymph? Are they transported in lipoprotein particles or bound to carrier proteins that are not captured in this type of analysis?
- 4) I was puzzled by their observation that "we observed normal membrane levels of PIP2/3 species in egr-expressing discs". If insulin PI3K signaling is reduced in these discs as assessed by increased nuclear FOXO, would you not expect the levels of PIP3 in the membrane to be reduced? Maybe it is difficult to see a decrease with this reagent. Please explain/clarify.
- 5) Can the authors please comment on the effect of some of their manipulations on the size of adult wings? In some ways that provides a good assay for the efficacy of regeneration (more than the quantification of stainings).
- 6) In the discussion, the authors should point out that this model of wounding is associated with an unusually high level of TNF/eiger because that is what is used to induce tissue death. Some of these changes might be less relevant in wounds that do not cause dramatic elevation in TNF levels.

At least some of the points that I have brought up suggest that the writing should be a little more careful with appropriate caveats pointed out.

Minor points (probably typos etc)

Line 141 "Fig 1N" I cannot find Fig. 1N

Line 151 Expression of either" Do you mean expression of eiger?

Figure 2C, D. What is "control". Are these larvae that have had the temperature shift but no eiger?

Figure 3E. What are the p values?

Reviewer #3

(Remarks to the Author)

The authors have nicely addressed the issues raised in my original review.

Version 2:

Reviewer comments:

Reviewer #1

(Remarks to the Author)

I appreciate the additional data and revisions. However, I remain unconvinced that the core conceptual and mechanistic

issues I raised in my previous reviews have been adequately addressed. Below, I highlight the most critical concerns that still require clarification or more rigorous treatment:

1. In my previous comments, I noted that the regions of ablation and knockdown in the Dual Control system were unclear. In response, the authors introduced demarcations that appear arbitrary. A more pressing concern is that Pdk1 or Stat92E inhibition appears to affect OPP signal even outside of the designated knockdown regions. The authors attribute this to a “broader systemic response to damage,” which they claim is described in detail in the manuscript. However, I was unable to find a clear explanation or analysis of this observation. This issue deserves more careful and quantitative evaluation, as it challenges the interpretation of tissue-autonomous effects.

2. A central weakness of the study remains unresolved — the mechanism by which JAK/STAT signaling regulates Pdk1 remains speculative. Despite raising this point in both of my previous reviews, the authors continue to dismiss it as “beyond the scope” of the study. However, given that this pathway forms the conceptual backbone of the manuscript, the absence of even preliminary mechanistic insight (e.g., transcriptional regulation, post-transcriptional control, or intermediary signaling) undermines the strength of the core claims. Without clarification of this regulatory link, the JAK/STAT–PDK1–TORC axis remains correlative and incomplete. At this stage, I am unable to support publication without further clarification

3. The manuscript proposes that Upd ligands (e.g., Upd3) reach peripheral tissues, yet these same tissues do not exhibit JAK/STAT or Pdk1 activation. This discrepancy is not adequately addressed. The authors suggest that context-dependent responses account for this divergence — a plausible idea, but one that remains speculative without spatially resolved evidence or functional validation. If the proposed model depends on selective activation of Pdk1 by Upd ligands, a clearer mechanistic rationale and supporting data are essential.

4. The authors argue that implicating Pdk1 upregulation in regenerative growth is itself a novel finding. I respectfully disagree. As I have consistently stated, without clarifying the mechanistic link between Upd signaling and Pdk1 regulation, the manuscript does not substantially advance current understanding in the field. In its present form, the findings remain primarily descriptive and offer only incremental novelty in the absence of mechanistic resolution.

Reviewer #2

(Remarks to the Author)

The authors have responded adequately to all of the points raised in my review.

Point by point response to reviewers

We sincerely appreciate the opportunity to revise our manuscript and thank you for your time and consideration. We are grateful for the insightful and constructive feedback from the reviewers, which has significantly strengthened our study. In response, we have extensively revised the manuscript to better highlight the novelty of each figure and have carefully addressed all concerns raised. Specifically, we have conducted new experiments (detailed below) to further support our key conclusions, restructured the manuscript to enhance the focus on the central role of PDK-1, and refined the writing to improve clarity. Additionally, we have incorporated new schematics, high-resolution images, and additional data to ensure a more comprehensive and compelling presentation of our findings. The review process also encouraged us to explore similarities in mechanisms between tissue damage and tumorigenesis. Before we address all concerns individually, we wanted to aid the review process and provide a brief summary of all new additional experimental data that was added to the revised manuscript.

New Figures	New experimental data
Fig 2G-N	Confirmation of peripheral tissue phenotypes using salm -GAL4 driver.
Fig 2R-T	Improving presentation and statistics for eye-antennal disc with rn -GAL4 driver.
Fig 3D and E	Triacylglyceride (TAG) quantification in fat body using Thin Layer Chromatography (TLC).
Fig 3M and N	TRE-RFP overlay with CG5535-GFP for spatial clarity.
Fig 3P-R	Improved data presentation.
Fig 4E-G	Additional experiment and quantification using an independent FOXO-GFP reporter line to visualize FOXO nuclear translocation. Improved data presentation of FOXO translocation.
Fig 4K and L	Alternative genetic ablation system using the pro-apoptotic gene hid .
Fig 4M and N	Spatial correlation between upregulation of Pdk1 and p-S6.
Fig 5L-N	Tissue specific knockdown of Pdk1 using DUAL Control genetic ablation system.
Fig 6A and B	Additional data to visualize spatial patterning of JNK/AP1 and JAK/STAT signaling.
Fig 6I-L	Genetic experiments for Pdk1 and JAK/STAT relationship.
Fig 6M-O	Heterozygous stat92E null background in egr -expressing larvae.
Fig 6P-R	Tissue-specific knockdown of stat92E using DUAL Control genetic ablation system.
Fig 7A-C	Spatial patterning of protein translation and Stat92E-dGFP in control and Ras, scrib tumor.
Fig 7D-F	Spatial patterning of Pdk1 expression and MMP1 (JNK) in control and Ras, scrib tumor.
Fig 7G-I	Protein translation in the peripheral notum tissue of control and Ras, scrib tumor-containing wing discs.
Fig 8A-C	Pdk1-GFP expression in the peripheral notum tissue of control and egr -expressing wing discs.
Fig 8F	Quantification of Pdk1-GFP expression in eye-antennal disc from control and egr -expressing larvae with improved statistical power.
Fig 8A-C	Pdk1-GFP expression in the peripheral notum tissue of control and Ras-scrib tumor.
Fig S2H and I	Improved visualization of the second mitotic wave (SMW) in eye-antennal discs using the rn -GAL4 driver, with enhanced image quality for better presentation of the data.
Fig S3.1B	Western blot analysis of Akt in fat body.
Fig S3.2	Improved presentation of single-cell RNA sequencing (scRNA-seq) data.
Fig S4L-Q	Alternative genetic ablation using pro-apoptotic gene reaper (rpr) and corresponding control for the hid system from Fig 4K and 4L.
Fig S5E and F	Improved data presentation of tGPH with an inset.
Fig S5G	Characterization of Pdk1-RNAi line.
Fig S6A and B	Spatial patterning of cell proliferation and Stat92E-GFP in control and Ras-scrib tumor.
Fig S6C-E	Control wing disc data for Fig 6C, 6E and 6G
Fig S6F and G	Undamaged wing disc controls for stat92E heterozygous null mutant.
Fig S7A	Spatial patterning of JNK/AP1 and JAK/STAT signaling in control and Ras-scrib tumor.
Fig S7B and C	Spatial patterning of protein translation and Stat92E-dGFP in control and Psc-Su(z)2 tumor.
Fig S8A and B	Pdk1-GFP expression in the peripheral notum tissue of control and DUAL Control egr -expressing wing discs.
Fig 8C-E	Quantification of Pdk1-GFP expression in leg disc from control and egr -expressing larvae with quantification.

Reviewer #1 (Remarks to the Author):

I view the manuscript as a loosely connected collection of potentially related data. The authors propose a mechanism based on correlative data without providing a clear demonstration of causality. Moreover, even if the data supports their claims, the conceptual advance presented in this manuscript appears incremental.

- Your criticism was heard and truly very much appreciated. In direct response, we performed additional experiments, restructured the manuscript and revised the focus of the figures. We hope that these changes clarify mechanistic insights and conceptual contributions of our work.

Major Issues:

1 A fundamental problem with this manuscript is the lack of genetic manipulation (except for the PDK1 heterozygous condition, which presents its own issues, as discussed later) alongside Eiger expression driven by rotund-GAL4, tub-GAL80ts. Since the development of the ablation system by the Hariharan lab at UC-Berkeley, independent manipulation of genes and ablation has been a challenge. However, the Harris lab at Arizona State University developed a system enabling simultaneous cell ablation and gene manipulation (Harris et al., eLife 2020). Without employing such a genetic approach, the authors failed to demonstrate a causal relationship among genes/proteins. Their claims of causality throughout the text are overstated.

We share the reviewer's concern regarding the tissue-specificity of our previous genetic manipulations. We have now included these **new experiments in the revised manuscript**:

- **Dual control system:** We have incorporated new experiments using the dual control system (Harris et al., eLife 2020) coupled with Eiger expression. We can demonstrate upregulation of OPP and PDK-1, as well as genetic dependency of proliferating cells on PDK-1 and STAT. The results are completely consistent with our model and the predicted epistasis (**Fig 5 L-N and Fig 6 P-R**).
- **General metabolic response:** In addition, we provide new data that demonstrates that if rn-GAL4, tub-GAL80ts drives expression of Rpr or Hid, we see elevated protein synthesis and PDK-1 expression in the regenerating domain. This indicates that the metabolic response we describe is not unique to Eiger-induced damage but is a general feature of different damage and regeneration systems (**Fig 4K, 4L and Fig S4L-U**).
- **Tumor pathologies:** Moreover, we provide new data obtained in the Ras scrib tumor model. Our data reveal a spatial separation between regions with high JAK/STAT, PDK-1 GFP, and OPP levels (proliferating cells) and regions with low levels of these markers (JNK-positive, slow-proliferating, senescent-like cells) (Fig 7 and Fig S7). Please note, that the mechanism mediating this spatial and functional separation of JNK and JAK/STAT signaling domains in regeneration and tumors were extensively described by us in (Jaiswal et al, 2023).

2 Except for the PDK1 results in Figs. 5-6, much of the data lacks novelty. The authors should more clearly distinguish their contributions from previously published work, particularly that of the Hariharan lab, which has reported many similar findings.

- Our initial draft focused on demonstrating that eiger discs induce all hallmarks of cachexia, which is maybe somewhat expected from published data on tumors and starvation, but has not been shown in all details for tissue damage, where the literature has focused less on the metabolic adaptations (some exceptions include: (Kashio & Miura, 2020; Kashio et al, 2016)) but more on mechanisms of developmental delay via ecdysone signaling (for example: (Colombani et al, 2012; Garelli et al, 2012; Hackney & Cherbas, 2014; Halme et al, 2010)) . Establishing the

systemic signaling and metabolic changes is an essential foundation to reveal our key and unexpected finding:

- **Novel physiological mechanism:** Proliferating cells can bypass inflammation-induced cachexia by cell-autonomously upregulating PDK-1. This renders them insulin-independent, enabling them to drive anabolic growth - while other tissues are locked in a catabolic state. Moreover, we now demonstrate that cachexia in the *egr*-model is due to insulin restriction due to low expression of Dilp2 and Dilp5 (**Fig 2C**). Insulin resistance may also play a role as we show that expression of *Impl2* from imaginal discs can act on fat body (**Fig 3F-J**).
- **Distinct from tumor models:** Tumors have long been known to induce cachexia and overcome it, the later by still little characterized mechanism. Our study demonstrates that even wild-type cells can overcome cachexia during regeneration. We offer a mechanisms via PDK-1 regulation (**Fig 4**), and reveal a physiological program for coordinating local requirement for anabolic growth with systemic catabolism during inflammation. We demonstrate that tumors induce the JAK/STAT-PDK1-S6 axis and thus conclude that tumors can highjack PDK-1 upregulation for their growth (see **Fig 7, S7**).
- **Differentiation from prior work:** While the Hariharan lab (and our own previous work) has extensively characterized the signaling environment and basic proliferative properties of these discs (e.g., through single-cell analyses as reported in (Floc'hlay et al, 2023; Worley et al, 2022)), none have addressed the metabolic reprogramming that enables wild-type cells to overcome cachexia. We thus respectfully disagree that the Hariharan lab has published similar data.

Manuscript revisions: We hope that our additional experiments and revised manuscript structure clarify mechanistic insights and conceptual contributions of our work.

3. There is a logical inconsistency regarding the relationship between OPP, EdU, p-S6, and PDK1. In Figure 1, there is no increase in OPP signals in proliferating cells, yet in later figures, the authors suggest that PDK1 enhances OPP signals. If PDK1 genuinely functions in regenerating cells, OPP should be upregulated in these cells.

- Here is where we think the logic inconsistency is perceived: We show that regenerating cells perform translation at levels higher than in peripheral tissues but comparable to undamaged discs (Figure 1). We argue that this is achieved by a relative upregulation of PDK-1 in regenerating cells, relative to wild type tissues and also relative to peripheral tissues (the latter exhibiting even reduced PDK-1 levels). So why do regenerating cells not also have even higher rates of translation than wild type cell, right?
- The important biological difference is: Maintenance of translation in Figure 1 occurs in an insulin-restricted environment where high levels of PDK-1 overcome low activity of canonical Insulin signaling pathway. This is central to understanding differences between Figure 1 and the following figures. In later figures, PDK1 overexpression in wild type discs occur in a normal insulin environment and leads to an overall increase in protein synthesis. This systemic difference explains the difference in how PDK-1 levels scale with levels of protein translation.
- Another difference is that the physiological levels of PDK-1 in regenerating cells may be lower than those generated by an UAS overexpression context. Especially, if PDK-1 levels are instructive for protein synthesis (which we think is the case, based on all data in our manuscript), then the observed differences may be influenced by different absolute PDK-1 levels.

Manuscript revisions: We have revised our manuscript text to strengthen the idea of PDK-1 being instructive for translation levels and restructured the figures to improve the logical flow of our argument. We hope these clarifications resolve the apparent inconsistency and clearly explain how PDK1 functions in different contexts.

4. The authors mention that Dilp8 or Eiger regulates insulin signaling, but then conclude

that ImpL2 is responsible for insulin reduction. How do they exclude the roles of Dilp8 and Eiger?

- To clarify, our data do not exclude potential roles for Dilp8 or Eiger in regulating systemic insulin signaling. In fact, we have explicitly cited literature that discusses their involvement in this process. Importantly, we now show that expression of Dilp2 and Dilp5 is downregulated in *egr*-expressing larvae. In the original draft we wanted to emphasize that many factors (Dilp8, Eiger, ImpL2, Upds and Mmp-1) are collectively expressed in eiger-discs and can thus contribute to a cachexic state, as described for tumors.
- With respect to ImpL2: We conclude that ImpL2 from the disc is sufficient to induce nutrient mobilization from the larval fat body, potentially by affecting insulin signaling in the fat body directly. We included this data, as ImpL2 was so far only shown to induce tumor cachexia in adults (Figuroa-Clarevega & Bilder, 2015). We have not observed that ImpL2 alters insulin signaling in imaginal discs (**see Reviewers' Figure below**) or other peripheral organs and thus do not conclude on a potential systemic effect.

Manuscript revisions: We now show that expression of Dilp2 and Dilp5 is downregulated in *egr*-expressing larvae. We now also clarify the distinction between the Dilp8 axis (IPCs) and the ImpL2 axis (fat body) in the text and included schematics in the figures.

5. In Figure 3, the relevance of much of the data is unclear. mTOR may be permissive rather than instructive in this context. For instance, even if they inhibit actin or tubulin, they might observe similar results. Moreover, the significant effect of rapamycin on the control condition is noteworthy. Additionally, the authors mention urea production, but they should be aware that the urea cycle in flies is defective due to the lack of ornithine transcarbamylase. This makes their claim misleading.

- Full S6K activation requires two phosphorylation events: one provided by mTORC and one provided by PDK-1. Our data confirm that mTORC is necessary and contributes to S6K activation in regenerating cells. We agree that mTORC act in a permissive capacity for S6K activation rather than being instructive per se. We think that our genetic experiments, as well as our observations related to PDK-1 regulation, underscore a more instructive role of PDK-1. To clarify these points, we have **revised the manuscript** with more precise terminology in our conclusions and restructured the figures.

- With respect to the urea cycle. Thank you! This was a genuine mistake on our end. We rewrote the section removing any speculation about the urea cycle.

6. It is unclear why the authors are showing STAT activation in the fat body in Figure 4. Additionally, they assess lipid droplets in the fat body, but they should measure TAG levels for a more accurate assessment.

- We provide **new experiments in the revised manuscript** to further characterize fat body changes. We measured triacylglyceride (TAG) levels in the fat body using TLC (**Fig 3D,E**). These experiments show that eiger larvae exhibit a decrease in fat body lipids - mirroring changes seen in starved larvae and tumor hosts. This reinforces our conclusion that the eiger discs induce nutrient mobilization from the fat body.
- Showing the data set for JAK/STAT activation in the fat body was intended to demonstrate that eiger-induced damage in the discs alters the systemic signaling environment (**Fig S3J-L**). In addition, STAT activation in the fat body has been associated with catabolic nutrient release for example (Ding et al, 2021; Hersperger et al, 2024; Shin et al, 2020). Although this dataset was well received by Reviewer #2, we have relocated these figures to the Supplement to maintain focus on our main message while still supporting the idea that inflammatory damage activates systemic pathways similar to those observed in tumor models.

7. In Figure 5, much of the data again appears to be merely correlative. Does STAT induce p-S6 through PDK1?

- We have performed **additional genetic experiments** that clarify the causal relationship between STAT signaling, PDK1 and translation: Reducing STAT levels by RNAi or STAT overexpression leads to a significant changes in PDK1 expression and OPP incorporation, which we now demonstrate occurs both during normal development as well as in regenerating cell populations. This indicates that STAT activity is required to maintain PDK1 levels and, consequently, protein synthesis (**Fig 6 and S6**).
- When STAT was overexpressed, we previously observed an upregulation of both PDK1, p-S6 and OPP. We struggled to obtain viable genotypes of STAT-OE and PDK-1RNAi expressing larvae and have therefore not been able to perform additional epistatic experiments. Yet we hope that the additional genetic experiments we performed, for example using the Dual control system, strengthen our conclusions sufficiently.

8. In Figure 6, the authors use PDK1 heterozygosity, which may affect JNK signaling in the ablated region, as PDK1 is known to regulate JNK. PDK1 heterozygosity could also influence other processes in distant organs, such as the fat body. As mentioned earlier, a tissue-specific approach is necessary.

- As discussed in our response to point 1, we have now included **new experiments in the revised manuscript** using a tissue-specific approach by utilizing the dual control system. These experiments, presented in **Fig 5 L-N and Fig 6P-R**, specifically target regenerating cells and convincingly clarify the cell-autonomous role of PDK1 (and STAT) in regeneration.
- Of note: In our PDK-1 heterozygosity experiments we did not observe appreciable changes to the activity of TRE-RFP, which suggests that PDK1 heterozygosity does not change JNK activity in the ablated region (**see Reviewers' Figure below**). Moreover, our literature search did not reveal evidence that PDK1 regulates JNK; rather, some mammalian studies suggest that JNK1 may modulate PDK1 activity.

9. The authors claim that other organs exhibit lower PDK1 levels, but in Figure 4, they demonstrate STAT activation in the fat body, which is mechanistically confusing.

- To clarify, our conclusions regarding PDK-1 downregulation exclusively refer to observations in peripheral imaginal discs and do not extend to the fat body.
- In the fat body, where we observe JAK/STAT activation, the functional outcome differs markedly from that in the discs. Specifically, while JAK/STAT promotes growth and anabolic processes in imaginal discs, its activation in the fat body is associated with nutrient mobilization and catabolism. This divergence is well documented in the literature, for example (Classen et al, 2009; Crucianelli et al, 2022; Ding et al., 2021; Hersperger et al., 2024; Jaiswal et al., 2023; Shin et al., 2020) . While we agree that this is confusing, it underscores how the same signaling pathway can regulate different effector networks in distinct tissues.

10. The high levels of PIP3, which is induced upon insulin binding to its receptor, in regenerating tissues suggest that the effects of FOXO are not due to Impl2, since Impl2 inhibits insulin's binding to its receptor. This contradicts their claim. If they think PI3K is activated not through insulin, they need to demonstrate how it occurs.

We would like to clarify our interpretation regarding PIP3 and FOXO in regenerating tissues:

- Insulin signaling and FOXO: The increased nuclear localization of FOXO in imaginal discs supports the notion that canonical insulin signaling is compromised in regenerating cells. We think that neither InR, PI3K or Akt are active (see Figure 3 and S3) . Importantly, we do not attribute this local increase of FOXO in regenerating cells to Impl2 secreted by JNK-signaling cells nearby, as we did not observe changes to Akt signaling in imaginal discs upon Impl2 overexpression (**see Reviewers' Figure below**). Our data on Impl2 only concludes on its effects in the fat body.
- PIP3 levels are maintained, not elevated: We do not claim that PIP3 is upregulated in regenerating cells. Rather, our observations indicate that PIP3 levels in rotund-GAL4, tub-GAL80ts, eiger discs remain comparable to those in control discs. This finding shows that the membrane recruitment of PDK1 via PIP3 - an essential step for PDK1 auto-activation - is still possible. Disclaimer: We currently do not know the precise mechanism of PIP3 generation in imaginal discs. Notably, expression of a dominant-negative InR in the rotund domain does not alter PH-GFP membrane recruitment. This observation suggests either that alternative receptor tyrosine kinases (RTKs) activate PI3K, creating saturating levels of PIP3, or that the PH-GFP

reporter may also (likely) detect PIP2 (see analysis of binding domains in (Britton *et al*, 2002; Lietzke *et al*, 2000)). Given that PDK1 can bind to both PIP2 and PIP3 (Komander *et al*, 2004; Levina *et al*, 2022), the precise necessity for PIP3 or PIP2 in our context remains uncertain. As a consequence, we have moved these supportive data to the supplement.

11. If their claim is correct, the novel aspect of their findings is the role of JAK-STAT and PDK1 in regenerating cells. It is therefore crucial to clarify how JAK-STAT regulates PDK1. Without this mechanistic insight, the contribution of this paper remains shallow.

- We appreciate the reviewer's focus on the mechanistic link between JAK-STAT signaling and PDK1 regulation. However, the major novel aspect of our study is that upregulation of PDK1 in regenerating cells enables them to bypass the systemic insulin restriction imposed by inflammatory damage. This allows these cells to sustain high levels of protein synthesis and rapid proliferation in an environment where other cells cannot. We hope that the restructuring of the paper clarifies this point.
- We provide new data to support the relationship between PDK-1 and JAK/STAT. Please refer to our responses to point 7, demonstrate that manipulation of JAK-STAT signaling alters PDK1 levels and, consequently, protein synthesis in regenerating cells. Moreover, the cited scRNA-seq data set (Floc'hlay *et al.*, 2023) suggest that this regulation is not mediated at the transcriptional level, implying that posttranslational mechanisms are likely involved. Unfortunately, a comprehensive dissection of these potential posttranslational mechanisms by which JAK-STAT regulates PDK1 is beyond the current scope of our study. Overall, we believe that our data establishes that PDK1 upregulation is the critical mechanism by which regenerating cells overcome a cachectic signaling environment. This functional insight, even without full mechanistic detail on the intermediate steps of JAK-STAT-mediated PDK1 stabilization, represents a significant advance in our understanding of regeneration under inflammatory conditions.

12. It would be ideal to demonstrate the spatial dynamics of Impl2 and UPD3, specifically whether they reach other tissues, such as the fat body.

- Based on the published literature and our phenotypes, we think that Impl2, Upd2 and Upd3 can act on distant organs such as the fat body (Ding *et al.*, 2021; Figueroa-Clarevega & Bilder, 2015; Kwon *et al*, 2015; Singh *et al*, 2025; Song *et al*, 2019). We discuss these factors to provide an idea of a mechanism for nutrient mobilization - as seen in tumor models and starvation. Given the extensive literature supporting their systemic roles, we respectfully ask the reviewer to reconsider the necessity of additional experiments on the spatial dynamics of Impl2 and Upd3. Instead, we have refocused our manuscript to emphasize that interorgan communication via

these factors provides the backdrop for our novel finding regarding PDK1's essential role in enabling regenerating cells to bypass insulin restriction.

13. Some figures, such as those showing FOXO and PIP3 imaging, require higher magnification for better clarity.

We have **performed new experiments** and provide the following improvements. We hope these changes address the reviewer's concerns regarding image clarity and data presentation.

- We validated our observations using a second, independent FOXO-GFP line, which confirms both the wild-type expression pattern and the spatial distribution observed in *eiger*-expressing discs (**Fig 2 D,E, Fig 4 E-G, Fig S4 A-C**).
- We acquired higher resolution images to ensure that key details are clearly visible, see also (**Fig S5 E,F**).
- We have added quantitative analyses of FOXO-GFP nuclear localization to further substantiate our findings.

14. The rotund-GAL4 driver promotes expression in the myoblasts beneath the notum. Some observations described in Figure 1 could be attributed to gene expression in myoblasts.

- We appreciate the reviewer's observation regarding the *rn-GAL4* expression in myoblasts. We had already included data showing that JNK activity in the notum is undetectable (**Fig S2A, B**), minimizing the concern that myoblast expression might influence the signaling environment. To further address this, **we have performed additional experiments**:
- We have now visualized also apoptosis in the notum (Fig S2A,B), as well as TRE-RFP and apoptosis in myoblasts (**see Reviewers' Figure below**). These experiments confirm that the signaling environment in the notum is not altered by *rn-GAL4* activity in the underlying myoblasts (under our conditions of 24 h *eiger* expression).

- We have confirmed our results using *salm-GAL4*, *tub-GAL80ts* - which expresses in the pouch and haltere, but not in the myoblasts, leg or eye disc (**see Reviewers' Figure below**). We now demonstrate that all peripheral phenotypes (i.e., reduced OPP incorporation, decreased EdU) upon *eiger* expression are highly reproducible (**Fig 2G-N**).

Collectively, data from these three independent genetic systems now show downregulation of translation, PDK-1 and proliferation in peripheral disc tissues. This evidence supports that our observations are not confounded by secondary expression of *rn-GAL4* in myoblasts.

15. In the text, they incorrectly refer to beta-gal expression, which should be beta-gal activity.

- Thank you, corrected.

Optional Point to Address:

In the introduction and discussion, the authors frame their findings in the context of tumorigenesis and tissue damage. Despite this, throughout the manuscript, they rely solely on *Eiger* expressed via *rotund-GAL4*, which limits the scope of their conclusions. While not essential to support their claims, it would be interesting to explore whether PDK1 functions in other contexts, such as *rpr*-mediated ablation and tumorigenesis.

As discussed already in our response to point 1, we have now included these **new experiments in manuscript revisions**:

- **Dual control system:** We have now incorporated new experiments using the dual control system (Harris et al., eLife 2020) coupled with *Eiger* expression. We can demonstrate upregulation of OPP and PDK-1, as well as genetic dependency of proliferating cells on PDK-1 and STAT. The results are completely consistent with our model and predicted epistasis (**Fig 5 L-N and Fig 6 P-R**).
- **Rpr, Hid systems:** In addition, we provide new data that demonstrates that if *rotund-GAL4*, *tub-GAL80ts* drives expression of *Rpr* or *Hid*, we similarly see elevated protein synthesis and PDK-1 expression in the regenerating domain. This indicates that the metabolic response we describe is

not unique to Eiger-induced damage but is a general feature of different damage and regeneration systems (**Fig 4K, 4L and Fig S4L-U**).

- **Tumor system:** Moreover, we provide new data in the Ras scrib tumor model. Our data reveal a spatial separation between regions with high JAK/STAT, PDK-1 GFP, and OPP levels (proliferating cells) and regions with low levels of these markers (JNK-positive, slow-proliferating, senescent-like cells) (**Fig 7, S7**). Please note, that the mechanism mediating this spatial and functional separation of JNK and JAK/STAT signaling domains in regeneration and tumors were extensively described by us in (Jaiswal *et al.*, 2023).

Reviewer #2 (Remarks to the Author):

In this manuscript the authors address the interesting observation that when imaginal discs are damaged, proliferation occurs right next to the wound to regenerate missing tissue but is suppressed in other parts of the wing disc. From their experiments they propose that proliferation is generally arrested because of reduced insulin signaling. However, in the proliferating region, JAK/STAT signaling activates Pdk1. Additionally the catabolism of the fat body results in increased circulating amino acids which promotes Tor activation in the regenerating tissue. Together, Pdk1 and Torc1 promote cell proliferation in this region.

This study addresses an important and interesting question - how proliferation is activated in one part of the tissue and suppressed in another. The authors make several interesting observations. However, there are issues of data presentation and data interpretation in the study as currently presented. Also, as discussed at the end of this review, the model proposed by the authors is not substantiated by the data presented – alternative interpretations are possible.

Detailed comments

Point 1

In Figure 1, the authors document the phenomenon nicely of proliferation in the eiger expressing domain and a suppression of proliferation in the rest of the disc. The authors present this as a completely novel discovery. However, the suppression of proliferation in non-regenerating portions of the disc was first shown by Sustar and Schubiger (2005) in transplantation experiments. Even for the eiger ablation system that the authors are using, the suppression of proliferation in non-regenerating parts of the disc was shown in multiple ways in the original paper that described the eiger ablation system (Smith-Bolton *et al* 2009).

- We apologize for not including these papers here and properly introducing this work to the reader. We do so now, and cite (Smith-Bolton *et al*, 2009; Sustar *et al*, 2011; Sustar & Schubiger, 2005).
- We would like to emphasize, that while these studies visualize low proliferation in the notum and antenna, in our understanding, neither study contextualized, quantified, nor mechanistically investigated this phenomenon, (nor did they directly compare proliferation to wild type disc).

Point 2

Figure 2 addresses the effects of tissue ablation in the wing disc on other remote tissues. There are however, many issues of data presentation and interpretation. In panels A and B the authors seem to think (line 697) that “the yellow arrowhead points to the expected location of the mitotic wave just anterior to the morphogenetic furrow” . This is incorrect. The arrowhead points to what is referred to historically as the “second mitotic wave” (SMW) which is a row of synchronous S-phases immediately posterior to the morphogenetic furrow. That only a portion of this row can be seen in panel A (it typically runs across the disc) indicates either that the preparation is not flat or that the EdU incorporation has not occurred through the entire preparation. The authors should familiarize themselves with the pattern of EdU incorporation in the normal eye disc. It would seem that the SMW is gone in panel B (if so, this is interesting), but this needs to be shown properly. Quantifying EdU incorporation in the yellow box does not really achieve anything clear because it includes the G1-arrested region in the furrow and the EdU incorporation anterior to the furrow (top right hand corner of the box). The eye disc also has different portions which are actively proliferating, paused in G1 and postmitotic. It is unclear which region of the eye disc is being measured in the OPP quantification in panel F. Each region in the wild-type should probably be compared to the corresponding region in the mutant condition.

We thank the reviewer for the detailed feedback on the eye disc. In response, we have undertaken the following actions:

- **Clarification of the mitotic wave identification:** We now refer to the region marked by the arrowhead correctly as the "second mitotic wave (SMW)" rather than the mitotic wave anterior to the furrow. Our revised figures and accompanying text have been updated to reflect this accurate terminology, ensuring consistency with historical literature (**Fig S2H, I**).
- **Refined Quantification:** We have **re-performed experiments** of EdU incorporation experiments in the eye disc. We co-stained with ELAV to clearly demarcate the different regions of the eye disc. We arrive at the same conclusions that proliferation in the SMW is reduced in the eiger-expressing condition. We have described our quantification approach in more detail for both EdU and OPP incorporation, describing quantification of comparable positions in wild type and rn-GAL4, tub-GAL80ts, eiger eye discs.

We hope these additional data sets address the reviewer’s concerns regarding data presentation and interpretation in Figure 2.

Point 3

The leg disc images seem more convincing– not sure why these data have not been quantified and have been relegated to a supplementary figure. One issue with this experiment is that rn-Gal4 is also expressed in the leg disc. The authors should check whether tissue is being killed in the leg disc. If so, this cannot be thought of as a response in tissue that is distant from the site of damage.

A similar point was raised by reviewer 1 (point 14) about expression of rn-GAL4, tub-GAL80ts, eiger in myoblasts. We have **performed the following experiments** to address the concerns:

- **Use of an alternative driver (salm-GAL4):** We have confirmed our results using salm-GAL4, tub-GAL80ts - which expresses in the pouch and haltere, but not in the myoblasts, leg or eye disc (**see Reviewers’ Figure for reviewer 1 point 14**). We now demonstrate that all peripheral phenotypes (i.e., reduced OPP incorporation, decreased EdU) upon eiger expression are highly reproducible (**Fig 2G-N**).

Point 4

In panels I, J and K, the authors claim that the levels of nuclear FOXO (as assessed by FOXO-GFP) in the fat body are increased when there is tissue ablation in the wing disc. Their FOXO-GFP is primarily localized in the nucleus. There are many published reports of FOXO staining (using an antibody) in the fat body in non-starved L3 larvae and FOXO is present throughout the cell (for example Figure 2 in Koyama et al. 2014). Thus, the FOXO-GFP reagent does not seem to accurately reflect FOXO localization and it is therefore questionable if these small differences in the intensity really reflect what is happening to FOXO itself.

We would like to clarify our findings and address the points raised as follows:

- We kindly ask the reviewer to consider that the reference. (Koyama et al., 2014) appears to describe the prothoracic gland rather than the fat body, where the cytoplasm/nucleus ratio is larger than in the fat body and FOXO is not expected to be so diluted in the cytoplasm. However, we agree that our observed differences in the fat body are small. We thus decided to remove this data and rely on **new data presented in Figure 2** to consolidate our evidence for systemic insulin restriction and the associated catabolic changes in the fat body.

Point 5

In Figure 3, the images showing the increase in P-S6 are convincing. The authors previous work suggests that the cells in the proliferating zone are proliferating faster. If this is the case, then the additional sensitivity to rapamycin might simply be because these cells are proliferating faster. Is the proliferating region also sensitive to inhibiting other growth pathways? Without testing that, it is hard to claim that the tor pathway is especially important for these cells.

- Our and other people's work has extensively characterized the distinct growth pathways activated in tissue damage models. Importantly, we compared the non-inflammatory hid system - which we described to rely on EGF/Ras/ERK and Hpo/Yki signaling - to the inflammatory eiger system, in which regenerating cells predominantly depend on JAK/STAT and Myc signaling (Crucianelli *et al.*, 2022; Jaiswal *et al.*, 2023; Smith-Bolton *et al.*, 2009). This work clearly established that JAK/STAT is a main driver of regenerative proliferation in the eiger system.
- Given that full activation of S6K requires two phosphorylation inputs -one from mTORC and one from PDK1 - it is logical to focus on these components. In our experiments, the sensitivity to Rapamycin in the proliferative zone is interpreted as a reflection of the necessity of mTORC for S6K activation. We revised the text to reflect this conclusion more clearly, and more strongly emphasize the central, and possible instructive role for PDK-1 in the subsequent figures.
- Of note: In mammalian systems, modulators such as JNK1, CDK5, and GSK-3 can influence S6K activity; however, in *Drosophila*, there is little known about such modulation.

To clarify these points, we have revised the manuscript to more clearly outline the history and rationale for our focus on JAK/STAT, mTORC and PDK1 as essential inputs for S6K activation.

Point 6

From the data in Panel L the authors conclude (line 253) that "Our results revealed an enrichment of many amino acids in egr-expressing hemolymph." But their data also show amino acids and peptides that are DECREASED in egr-expressing hemolymph (Asp Lys, Asn Gln Lys, Glu Ile Arg). Thus the data presented do not seem consistent with a general increase in amino acid levels in the hemolymph when eiger is expressed in the wing disc.

- We agree that not all amino acids are uniformly increased in the hemolymph of *eiger*-expressing larvae. Instead, our data reveal a distinct amino acid signature characterized by relative enrichment of certain amino acids, while others are decreased. We have rephrased the relevant section (line 253) to emphasize that our findings reflect a specific alteration in the amino acid profile, rather than a global increase in all amino acid levels.

Point 7

In Figure 4 The authors show nicely that there is increased JAK/STAT signaling in the fat body.

Two things I find confusing about the data shown in this experiment:

- 1) In panels A and B, both the size and roundness of lipid droplets increases in the *eiger* condition. What do the authors mean by roundness? Please explain.
- 2) If the lipid droplets are larger in the *eiger* condition, how is this consistent with their hypothesis that nutrients are being mobilized from the fat body to fuel the regeneration. Would this not lead to a depletion of triglycerides in the fat body? Please explain.

- "Roundness" refers to the mathematical relationship between the major axis length and perimeter of lipid droplets, a quantifiable metric used to assess droplet morphology. In the literature (e.g., (Dark *et al*, 2022; Figueroa-Clarevega & Bilder, 2015; Gutierrez *et al*, 2007)) - an increase in roundness (i.e., a smoother, more circular appearance) is observed during fat body mobilization, such as under starvation or tumor conditions. We interpret this as an early molecular change in lipid storage dynamics that even precedes a reduction in overall droplet size. This either reflects changes in lipid droplet subpopulations (Ugrankar *et al*, 2019), or molecular droplet dynamics (Beller *et al*, 2010) during mobilization.
- Please also consider our response to **Reviewer 3, point 5**.
- To more robustly assess nutrient mobilization, we now also provide **new experimental data** where we measured triacylglyceride (TAG) levels in the fat body using TLC (**Fig 3D,E**). This experiment shows that *eiger* larvae exhibit a decrease in fat body TGAs, mirroring changes seen in starved larvae and tumor hosts. These data support our hypothesis that nutrients are mobilized from the fat body to fuel regeneration.

Point 8

In Figure 5 the authors claim that “both cytoplasmic and nuclear levels of FOXO were elevated”. How can we see this in the low magnification images shown? Also note the earlier point of how FOXO-GFP may not accurately reflect FOXO localization.

The data showing Pdk1-GFP is convincing. (It would be nice if the authors stated clearly if the GFP is a transcriptional reporter or a protein fusion in each case that they use these reagents).

In response, we provide the following revisions and clarifications:

- Validation with a second FOXO-GFP line: We validated our observations using a second, independent FOXO-GFP line that has been previously published. This line confirms both the wild-type expression pattern and the spatial distribution observed in *eiger*-expressing discs, reinforcing our conclusions.
- Improved imaging and resolution: We acquired higher-resolution images to ensure that key details of FOXO localization are clearly visible. Given that the cytoplasm of imaginal disc cells is small, our analysis primarily focuses on nuclear FOXO, which should be the principal determinant of FOXO activity. We have clarified in the revised text that our conclusions are drawn mainly from the quantification of nuclear FOXO signal.
- Characterization of the PDK-1 GFP line: We now give more information about the PDK-1 GFP line in the manuscript (see materials and Methods): The Pdk1-GFP line (BDSC: 59836) was characterized using the following approaches: a complex expression pattern can be observed in wild type tissues in immunofluorescence, where GFP is detected in the cytoplasm but also to

apical-junctional membranes. Tissues from larvae produce a band of the expected size on Western blot. Three independent RNAi lines (BDSC: 27725, BDSC: 34936, BDSC: 36071) targeting the 3' region of the PDK-1 transcript downstream of the GFP-cassette insertion site can robustly knock-down GFP expression. We conclude that a full-length, membrane-recruitable and genetically tractable protein is produced from the endogenous locus and that the annotation of the insertion site in FlyBase is incorrectly oriented...

Point 9

The data shown in Figure 6 are convincing – that Pdk1 promotes growth and cell cycle progression, and that regenerative proliferation is sensitive to Pdk1 levels.

In summary, the authors do show that the JAK/STAT pathway and Pdk1 are important for the proliferation of cells around the eiger-damaged tissue. Since they have not looked at other pathways (e.g. RTK, Myc, Hippo) I don't see how they can conclude that this pathway is especially important – similar results may be observed if other pathways are tested. Also if the JAK/STAT ligands can reach distant sites such as the fat body, why is this pathway not getting activated in other parts of the wing disc such as the notum which is much closer to the site of release. Why is this pathway only operating in the proliferating zone?

- We address concerns about other growth pathway in our response to point 5 above.
- With respect to JAK/STAT activation patterns: The spatial pattern of JAK/STAT activation in the imaginal disc is highly restricted even during normal development. Studies have shown that JAK/STAT ligands such as Upd1 are produced in discrete domains, with JAK/STAT activation confined largely to regions in close proximity to the Upd source (Ayala-Camargo et al, 2013; Johnstone et al, 2013). Additionally, cell-specific factors likely modulate signaling 'receptiveness' and 'competence' (Fisher et al, 2016; Jaiswal et al., 2023; Stec et al, 2013; Vidal et al, 2010). For example, we previously found that JAK/STAT activation is suppressed in JNK-signaling cells which actually produce Upd ligands, thereby contributing to the spatial segregation of signaling in eiger discs (Jaiswal et al., 2023). Although the three ligands Upd1,2,3 may 'travel' different distances (Wright et al, 2011), these regulatory mechanisms likely create specific pathway activation patterns. But yes, we have yet to explain why specifically only the proliferative zone but not peripheral disc cells activate it.

Point 10

The data presented also have many technical issues – especially Figures 3 -5. These issues need to be sorted out.

The data presented in the UMAPs in supplemental figure S3 need to be explained better. From the figure legend it seems that the points represent cells from both control and egr-expressing discs. Without showing the two conditions separately, how do we know whether these expressing cells are from control or eiger-expressing discs? Also ImpL2 seems to be expressed most highly in a part of the disc that is very different from where all the wound ligands are expressed. What is happening with ImpL2?

- We have generated and now present separate UMAP plots for control and eiger-expressing discs.
- We have not investigated why there is an additional ImpL2-expressing cell population outside the wound cluster which expresses the transcript.

Point 11

Another point is that the referencing of the literature needs to be improved. The authors cite their own work a lot but do not cite previous work that has made the same observations. This is observed in multiple places in the manuscript. For example (line 191) "We previously

reported that *egr*-expressing larvae also exhibit a significant delay in initiating pupariation (La Fortezza et al 2016)”. This observation has a long history. The delay following tissue damage from radiation was first described by Hussey (1927) and from damage to imaginal discs by Simpson et al (1980). Even the first description of the pupariation delay after damaging imaginal discs by *eiger* expression was first described by Smith-Bolton et al. (2009) - long before La Fortezza et al.

- We thank the reviewer for highlighting the need to better contextualize our findings with the long-standing literature on this subject. We apologize for not having included these references in the original manuscript. To slim down on peripheral data, we removed the description of the delay from the manuscript. We hope that this is ok.

Reviewer #3 (Remarks to the Author):

This is an interesting study that looks at how metabolism of proximal and distal tissues is altered when a wound is inflicted in a tissue. Interestingly, it shows that the regions proximal to the wound are able to continue proliferating and growing due to up-regulation of PDK1 expression, despite a likely systemic drop in insulin signaling. This study will be interesting for a broad audience studying growth, insulin/TOR signaling, tissue damage and regeneration, and translation. Overall the data are of good quality. I only have minor issues as suggestions for improving the manuscript.

Minor Issues

1. Suppl. Fig. 1D-F: It would be helpful to quantify the proportion of cells that are in the different phases of the cell cycle in the control pouch and in the *rn>egr* region

- We have previously quantified the proportion of cells in different cell cycle phases in both the control the *rn-GAL4*, *tub-GAL80ts*, *eiger* discs, as reported in our earlier work (Cosolo et al, 2019; Crucianelli et al., 2022). Accordingly, we have cited these studies in the manuscript, and hope they sufficiently support our conclusions regarding cell cycle dynamics in these regions.

2. Line 146 says “They induce compensatory proliferation in surrounding cells, which can be detected by rapid EdU incorporation (Fig 1A,B)”

To me the EdU does not look higher around the wound in panel B compared to the control disc in panel A ? Either quantify or rephrase the sentence?

Or is it rather that proliferation is inhibited in the rest of the *rn>egr* disc ?

- We have previously characterized this effect extensively (Crucianelli et al., 2022). To create a better focus in the paper we deemphasize this effect and rather highlight that that these cells continue to proliferate even when surrounding cells do not.

3. It would be good to solidify that there’s reduced systemic insulin signaling in animals with wing lesions compared to controls. E.g. do a WB on pAkt on salivary glands or fat body or whole larvae?

We thank the reviewer for the suggestions and in our revised manuscript, we have incorporated additional data as follows:

- We have also added new data in **Figure 2 and S2** reinforcing our interpretation of a reduced systemic insulin signaling state in *eiger*-expressing larvae. For example:
 - Expression of Dilp2 and Dilp5 is downregulated in *eiger*-expressing larvae, similar to starved larvae, confirming an insulin-restricted environment.
 - We show that FOXO undergoes nuclear translocation in peripheral imaginal disc tissue of *rn-GAL4*, *tub-GAL80ts*, *eiger* larvae, confirming that systemically canonical Insulin signaling is low.
 - We performed Western blots on fat body tissue, probing for both total Akt and phosphorylated Akt (p-Akt). Our results show that p-Akt is nearly undetectable in the fat bodies of *rn-GAL4*, *tub-GAL80ts*, *eiger* larvae – and comparable to levels seen in starved larvae. Although total Akt levels are also low, these observations suggest that the activity of the Akt pathway is significantly reduced under *eiger*-expressing conditions. This effect was consistently observed in three independent biological experiments.

4 Fig 3I-K: I see clearly the upregulation of CG5535 in the senescent domain, but it's not so visually obvious in the proliferative domain. How is the senescent domain marked, to make sure it's not getting included in the quantification of the proliferative domain? Could it be that the mild increase in CG5535-GFP shown in the quantification in panel K is actually coming from the senescent domain? Same for Fig S3E-F.

To address these concerns, we have made the following revisions:

- We have now included the TRE-RFP channel in our CG5535-GFP staining (**Fig 3 M-O**). This marker clearly delineates the JNK-signaling (senescent) domain from the proliferative region. Consequently, our quantification of CG5535-GFP is confined exclusively to the proliferative domain (remember, defined as a 20 μ m band outside the JNK domain). Although there is an upregulation of CG5535 in the senescent domain, also the proliferative region shows a significant increase.
- While our presentation always suggests a bistable scenario (senescent versus proliferating), the actual response in the tissue is more graded within and around the *eiger*-expression domain. There are differences in gradient range among responding genes (see **Fig S3.2 C**) but they are, at present, beyond the scope of our experimental resolution.

5 Fig 4A-D, the authors observe some morphological changes to lipid droplets in the fat body. They appear to actually get larger and stain more intensely with Nile red. They then conclude “the observed hemolymph changes may be derived from fat body catabolism”. I would rather guess these changes look like lipid anabolism, rather than catabolism? But the “may” is critical because I suppose it is possible. But the ‘may’ then disappears from the rest of the manuscript, as this conclusion gets taken for granted. For instance, at the end of the next paragraph: “These observations indicate that the production of Impl2 and Upd cytokines from *egr*-expressing discs are likely responsible to drive nutrient mobilization from the fat body...” If the conclusion that the fat body mobilizes nutrients is important, it should be shown solidly. Otherwise, the respective sentences should be rephrased, also in the discussion.

A related point was also raised by Reviewer 2. We agree that the observed morphological changes to lipid droplets—in particular, the apparent increase in droplet size—might seem counterintuitive when considering nutrient mobilization. However, our interpretation is supported by both the literature and our new experimental data, as detailed below:

- "Roundness" refers to the mathematical relationship between the major axis length and perimeter of lipid droplets, a quantifiable metric used to assess droplet morphology. In the literature (e.g., (Figueroa-Clarevega & Bilder, 2015; Gutierrez *et al.*, 2007)) - an increase in roundness (i.e., a smoother, more circular appearance) is observed during fat body mobilization, such as under starvation or tumor conditions. We interpret this as an early molecular change in lipid storage dynamics that even precedes a reduction in overall droplet size. This either reflects changes in lipid droplet subpopulations (Ugrankar *et al.*, 2019), or molecular droplet dynamics (Beller *et al.*, 2010) during mobilization.
- To follow up on this apparent change in size distribution, we performed additional experiments comparing fat body lipid droplets under three conditions: eiger expression, scrib tumor formation, and starvation. We observed that all conditions—representing increasing levels of a cachexic, catabolic signaling environment (from tissue damage < tumor < starvation)—display an increase in lipid droplet roundness and a loss of small lipid droplets, which may account for the perceived increase in size. At higher levels of cachexia (in tumors and under starvation), there is also a reduction in overall droplet size. See **Reviewers' Figure** below.

- To more robustly assess nutrient mobilization, we now also provide new experimental data where we measured triacylglyceride (TAG) levels in the fat body using TLC (Figure 2). Eiger larvae exhibit a decrease in fat body TGA - mirroring changes seen in starved larvae and tumor hosts. We hope these additional data and clarifications satisfactorily address the reviewer's concerns regarding the interpretation of lipid droplet morphology and nutrient mobilization.

6. Line 308: "If egr-expressing discs induce low systemic Insulin signaling..."

It would be good to rephrase this sentence because "induce" usually means to increase, so it sounds like this sentence is saying that insulin signaling is going up rather than down. Perhaps "cause" instead of "induce"?

- Thank you. Corrected.

7 One major conclusion is that the proliferative zone is able to continue proliferating because JAK/STAT signaling induces PDK1 expression. Can this be shown epistatically? ie if JAK/STAT signaling is inhibited (e.g. with loss-of-function mitotic clones) in the proliferative area, does this block PDK1 upregulation? (I don't know if this experiment is logistically possible.)

- We have also performed **additional genetic experiments** that clarify the causal relationship between STAT signaling, PDK1 and translation: Reducing STAT levels by RNAi leads to a significant decrease in both PDK1 expression and OPP incorporation, which we now demonstrate occurs both during normal development as well as in regenerating cell populations. This indicates that STAT activity is required to maintain PDK1 levels and, consequently, protein synthesis. When STAT was overexpressed, we previously observed an upregulation of both PDK1, p-S6 and OPP. We struggled to obtain viable genotypes of STAT-OE and PDK1-RNAi expressing larvae and have therefore not been able to perform additional epistatic experiments. Yet we hope that the additional genetic experiments we performed, for example using the Dual control system, strengthen our conclusions sufficiently.

References

Ayala-Camargo A, Anderson AM, Amoyel M, Rodrigues AB, Flaherty MS, Bach EA (2013) JAK/STAT signaling is required for hinge growth and patterning in the Drosophila wing disc. *Dev Biol* 382: 413-426

Beller M, Bulankina AV, Hsiao HH, Urlaub H, Jackle H, Kuhnlein RP (2010) PERILIPIN-dependent control of lipid droplet structure and fat storage in Drosophila. *Cell Metab* 12: 521-532

Britton JS, Lockwood WK, Li L, Cohen SM, Edgar BA (2002) Drosophila's insulin/PI3-kinase pathway coordinates cellular metabolism with nutritional conditions. *Dev Cell* 2: 239-249

Classen AK, Bunker BD, Harvey KF, Vaccari T, Bilder D (2009) A tumor suppressor activity of Drosophila Polycomb genes mediated by JAK-STAT signaling. *Nat Genet* 41: 1150-1155

Colombani J, Andersen DS, Leopold P (2012) Secreted peptide Dilp8 coordinates Drosophila tissue growth with developmental timing. *Science* 336: 582-585

Cosolo A, Jaiswal J, Csordas G, Grass I, Uhlirva M, Classen AK (2019) JNK-dependent cell cycle stalling in G2 promotes survival and senescence-like phenotypes in tissue stress. *Elife* 8

Crucianelli C, Jaiswal J, Vijayakumar Maya A, Nogay L, Cosolo A, Grass I, Classen AK (2022) Distinct signaling signatures drive compensatory proliferation via S-phase acceleration. *PLoS Genet* 18: e1010516

Dark C, Cheung S, Cheng LY (2022) Analyzing cachectic phenotypes in the muscle and fat body of Drosophila larvae. *STAR Protoc* 3: 101230

Ding G, Xiang X, Hu Y, Xiao G, Chen Y, Binari R, Comjean A, Li J, Rushworth E, Fu Z *et al* (2021) Coordination of tumor growth and host wasting by tumor-derived Upd3. *Cell Rep* 36: 109553

Figuerola-Clarevega A, Bilder D (2015) Malignant Drosophila tumors interrupt insulin signaling to induce cachexia-like wasting. *Dev Cell* 33: 47-55

Fisher KH, Stec W, Brown S, Zeidler MP (2016) Mechanisms of JAK/STAT pathway negative regulation by the short coreceptor Eye Transformer/Latran. *Mol Biol Cell* 27: 434-441

Floc'hlay S, Balaji R, Stankovic D, Christiaens VM, Bravo Gonzalez-Blas C, De Winter S, Hulselmans GJ, De Waegeneer M, Quan X, Koldere D *et al* (2023) Shared enhancer gene regulatory networks between wound and oncogenic programs. *Elife* 12

Garelli A, Gontijo AM, Miguela V, Caparros E, Dominguez M (2012) Imaginal discs secrete insulin-like peptide 8 to mediate plasticity of growth and maturation. *Science* 336: 579-582

Gutierrez E, Wiggins D, Fielding B, Gould AP (2007) Specialized hepatocyte-like cells regulate *Drosophila* lipid metabolism. *Nature* 445: 275-280

Hackney JF, Cherbas P (2014) Injury response checkpoint and developmental timing in insects. *Fly (Austin)* 8: 226-231

Halme A, Cheng M, Hariharan IK (2010) Retinoids regulate a developmental checkpoint for tissue regeneration in *Drosophila*. *Curr Biol* 20: 458-463

Hersperger F, Meyring T, Weber P, Chhatbar C, Monaco G, Dionne MS, Paeschke K, Prinz M, Gross O, Classen AK *et al* (2024) DNA damage signaling in *Drosophila* macrophages modulates systemic cytokine levels in response to oxidative stress. *Elife* 12

Jaiswal J, Egert J, Engesser R, Peyroton AA, Nogay L, Weichselberger V, Crucianelli C, Grass I, Kreutz C, Timmer J *et al* (2023) Mutual repression between JNK/AP-1 and JAK/STAT stratifies senescent and proliferative cell behaviors during tissue regeneration. *PLoS Biol* 21: e3001665

Johnstone K, Wells RE, Strutt D, Zeidler MP (2013) Localised JAK/STAT pathway activation is required for *Drosophila* wing hinge development. *PLoS One* 8: e65076

Kashio S, Miura M (2020) Kynurenine Metabolism in the Fat Body Non-autonomously Regulates Imaginal Disc Repair in *Drosophila*. *iScience* 23: 101738

Kashio S, Obata F, Zhang L, Katsuyama T, Chihara T, Miura M (2016) Tissue nonautonomous effects of fat body methionine metabolism on imaginal disc repair in *Drosophila*. *Proc Natl Acad Sci U S A* 113: 1835-1840

Komander D, Fairservice A, Deak M, Kular GS, Prescott AR, Peter Downes C, Safrany ST, Alessi DR, van Aalten DM (2004) Structural insights into the regulation of PDK1 by phosphoinositides and inositol phosphates. *EMBO J* 23: 3918-3928

Kwon Y, Song W, Droujinine IA, Hu Y, Asara JM, Perrimon N (2015) Systemic organ wasting induced by localized expression of the secreted insulin/IGF antagonist Impl2. *Dev Cell* 33: 36-46

Levina A, Fleming KD, Burke JE, Leonard TA (2022) Activation of the essential kinase PDK1 by phosphoinositide-driven trans-autophosphorylation. *Nat Commun* 13: 1874

Lietzke SE, Bose S, Cronin T, Klarlund J, Chawla A, Czech MP, Lambright DG (2000) Structural basis of 3-phosphoinositide recognition by pleckstrin homology domains. *Mol Cell* 6: 385-394

Shin M, Cha N, Koranteng F, Cho B, Shim J (2020) Subpopulation of Macrophage-Like Plasmacytes Attenuates Systemic Growth via JAK/STAT in the *Drosophila* Fat Body. *Front Immunol* 11: 63

Singh A, Hu Y, Lopes RF, Lane L, Woldemichael H, Xu C, Udeshi ND, Carr SA, Perrimon N (2025) Cell-death induced immune response and coagulopathy promote cachexia in *Drosophila*. *bioRxiv*

Smith-Bolton RK, Worley MI, Kanda H, Hariharan IK (2009) Regenerative growth in *Drosophila* imaginal discs is regulated by Wingless and Myc. *Dev Cell* 16: 797-809

Song W, Kir S, Hong S, Hu Y, Wang X, Binari R, Tang HW, Chung V, Banks AS, Spiegelman B *et al* (2019) Tumor-Derived Ligands Trigger Tumor Growth and Host Wasting via Differential MEK Activation. *Dev Cell* 48: 277-286 e276

Stec W, Vidal O, Zeidler MP (2013) *Drosophila* SOCS36E negatively regulates JAK/STAT pathway signaling via two separable mechanisms. *Mol Biol Cell* 24: 3000-3009

Sustar A, Bonvin M, Schubiger M, Schubiger G (2011) *Drosophila* twin spot clones reveal cell division dynamics in regenerating imaginal discs. *Dev Biol* 356: 576-587

Sustar A, Schubiger G (2005) A transient cell cycle shift in *Drosophila* imaginal disc cells precedes multipotency. *Cell* 120: 383-393

Ugrankar R, Bowerman J, Hariri H, Chandra M, Chen K, Bossanyi MF, Datta S, Rogers S, Eckert KM, Vale G *et al* (2019) *Drosophila* Snazarus Regulates a Lipid Droplet Population at Plasma Membrane-Droplet Contacts in Adipocytes. *Dev Cell* 50: 557-572 e555

Vidal OM, Stec W, Bausek N, Smythe E, Zeidler MP (2010) Negative regulation of *Drosophila* JAK-STAT signalling by endocytic trafficking. *J Cell Sci* 123: 3457-3466

Worley MI, Everetts NJ, Yasutomi R, Chang RJ, Saretha S, Yosef N, Hariharan IK (2022) Ets21C sustains a pro-regenerative transcriptional program in blastema cells of *Drosophila* imaginal discs. *Curr Biol* 32: 3350-3364 e3356

Wright VM, Vogt KL, Smythe E, Zeidler MP (2011) Differential activities of the *Drosophila* JAK/STAT pathway ligands Upd, Upd2 and Upd3. *Cell Signal* 23: 920-927

Revisions

We sincerely thank all of the reviewers for their thoughtful and constructive comments. We're pleased to see that we were able to successfully address many of the points raised. In response to the current feedback, we have made the following revisions to the manuscript:

Figure	Changes in figure (and associated figure legends)
Fig. 2D and 2E	Included panels D' and E' to show magnified images of dFOXO-GFP localization.
Fig. 5L and 5M, Fig. 6P and 6Q, Fig. S6H and S6I	Included spatial markers for the ablated region and the dve-GAL4 expressing domain to better visualize the knockdown and ablated areas. Additionally, we included DAPI channels and MMP1 staining to better demarcate the ablated and regenerating regions.
Fig. S5E and S5F	Removed. See discussion in our responses to Reviewer 2.

In addition, we have made minor wording adjustments throughout the text to better reflect more cautious conclusions and limitations of the study.

Reviewer #1 (Remarks to the Author):

The authors have addressed some of my previous comments; however, they have failed to address the most fundamental issues raised in my original review.

1. In my initial review, I pointed out that one of the core problems was that PDK involvement was shown only in a correlative manner. In response, the authors introduced the Dual Control approach. Unfortunately, the data presented are not convincing. Figures 5L–M and 6P–Q are particularly problematic and lack clarity. Notably, even regions outside of the wing pouch exhibit reduced signal, making it difficult to interpret the results. The authors need to clarify which region is ablated and which region expresses RNAi. Without such demarcation, the conclusions drawn are unreliable.

Response:

We appreciate the reviewer's concern regarding the spatial demarcation of gene knockdown in the Dual Control system. To improve clarity in Figures 5L–M and 6P–Q, we have now included spatial demarcations that distinguish the damage site from the surrounding region where the knockdown of either *Pdk1* or *STAT92E* occurs:

- The tissue domain expressing *eiger* driven by a *salM* promoter in the central pouch was always identified by co-staining for the JNK target gene MMP-1. We include now the MMP-1 channel in the updated Figures 5.
- The tissue domain expressing *Pdk1* RNAi driven by *dve-GAL4* in the pouch was identified by morphological landmarks of the pouch folds that were determined by co-expression of UAS-GFP in initial experiments and as characterized in Fig 3 G-K in (Harris et al., 2020).

With this updated visualization, we hope to better emphasize the effects observed within *Pdk1* RNAi expressing cells. The observed reduction in translation outside the wing pouch reflects the broader systemic response to damage, which we have described in detail in the manuscript.

11. The authors did not address the key mechanistic question of how Stat induces PDK. This remains the most critical unresolved issue in the current manuscript.

Response:

While this is indeed an important aspect of the regenerative signaling network, the focus of our current study is specifically on how regenerating cells overcome systemic reduction in insulin signaling by upregulating Pdk1 and sustaining TORC activity. We believe this insight in itself constitutes a significant advance in understanding tissue regeneration or tumor growth. Addressing post-translational mechanisms underlying JAK-STAT-mediated regulation of Pdk1 will require a large number of biochemical and functional experiments and broader screening approaches, which we plan to pursue in future studies.

12. I had requested clarification of the spatial dynamics of Upd3, which is crucial to their hypothesis. The authors speculate that Upd ligands activate PDK in proliferating tissues such as the wing disc and also reach peripheral tissues like the fat body. However, they also claim that PDK is suppressed in these peripheral tissues. This is logically inconsistent. The dynamics of Upd ligands (comment 12) and their relationship with PDK activation (comment 11) are essential for supporting the central claim of this study. Without addressing these points, the manuscript offers only marginal novelty.

Response:

We previously articulated our position. However, we would like to take this opportunity to highlight the key points

- We focus on **insulin-Pdk1-protein synthesis** axis. The link to JAK/STAT activation ties our findings into the extensive literature that JAK/STAT activation per se is important for compensatory proliferation and tumor growth.
- The properties of individual Upd ligands and how exactly JAK/STAT is activated in imaginal disc, muscles, fat bodies, tumors, regenerating tissue or other organs by Upd1, Upd2 or Upd3 is beyond the scope of this study. A body of literature has tried to address if and how Upd3 reaches distant organs, revealing the **limited experimental power** we currently have to follow the precise spatial dynamics of the Upd3 protein itself (Ding et al., 2021, Hersperger et al., 2024, Jiang et al., 2009, Romao et al., 2021, Shin et al., 2020, Wright et al., 2011). Our study does thus not aim to solve the (and we absolutely agree) important open question, why JAK/STAT is only activated in the regenerating domain and in the fat body, even though Upd1, Upd2 or Upd3 may be able to also reach peripheral Imaginal disc.
- In the fat body, where we observe JAK/STAT activation, the functional outcome differs significantly from that in the damaged imaginal discs. While JAK/STAT promotes growth and anabolic processes in imaginal discs, its activation in the fat body is associated with nutrient mobilization and catabolic activity. This divergence is well documented in previous studies (Classen et al., 2009; Crucianelli et al., 2022; Ding et al., 2021; Hersperger et al., 2024; Jaiswal et al., 2023; Shin et al., 2020). Although this contrast may appear confusing, it highlights how the **same signaling pathway can regulate distinct effector networks depending on organ context (but may thereby coordinate both proliferation and nutrient supply)**.
- We never claim that Pdk1 is downregulated in the fat body. We only claim that Pdk1 is downregulated in peripheral imaginal discs. This would be consistent with a model where Pdk1 is positively regulated in this actively growing, proliferating tissue not just by JAK/STAT signaling, but also by Insulin signaling in a positive feedback loop. **Since peripheral imaginal discs have neither JAK/STAT signaling nor Insulin signaling in egr-expressing larvae, Pdk1 levels are reduced.** However, proving this model is an entirely new project in the lab and it would distract from the take home message of the paper.

13. I requested higher-magnification images of FOXO-GFP (Fig. 2D–E, etc.) and PIP3, but the authors did not provide them. Higher magnification is necessary for readers to properly assess the data. The current images are insufficient; figures should be interpretable without relying solely on quantitative plots.

Response:

We have included yet even higher-magnification images for the dFOXO-GFP data presented in Figures 2D–E and 4E–F to provide greater visual clarity and detail.

Lastly, several points made by the authors in the rebuttal should be explicitly reflected in the revised manuscript text.

Response:

We have made additional changes to reflect more balanced arguments and conclusions in the now revised results and discussion. We hope that these changes address some of the reviewers' concerns.

Reviewer #2 (Remarks to the Author):

The revised manuscript is much improved. The authors have addressed most of my concerns. This now makes a nice story that is supported adequately by the data. The data are well presented. I still have some issues that can be addressed mostly by textual changes. These are:

- 1) The authors should clarify the nature of the Pdk1-GFP reagent. I assume this is construct that has a GFP fused to the endogenous Pdk1. Is this correct? Or is it simply a transcriptional reporter? It is important to explain this clearly. Otherwise it is hard to know whether it reflects Pdk1 transcription, Pdk1 protein levels, or both. I was not able to find this explanation anywhere.

Response:

Thank you for pointing this out. While we had described the Pdk1-GFP fly line and its characterization in the Materials and Methods section of the revised manuscript, we had not referenced this in the Results section where Pdk1 is first introduced. We have now added a reference to the relevant subsection in Materials and Methods at the point where Pdk1-GFP is first mentioned in the Results, to guide readers to the details.

- 2) I think the conclusion that Pdk1 is sufficient for regenerative proliferation (line 283) is overstated. The wound region makes many factors that likely function together with Pdk1 to promote regenerative growth. Sufficiency implies that it can do it on its own.

Response:

Yes, we completely agree and apologize for the choice of words, including the oversimplified heading of this section (Figure 5). We have revised the heading and the text to more accurately reflect the data and maintain a balanced interpretation of our findings.

- 3) The authors conclude that the increase in some amino acids in the hemolymph is a result of their mobilization from the fat body. This is likely but other explanations are formally possible. The authors show lipolysis in the fat body and an increase in amino acids in the hemolymph. The amino acids could be coming from somewhere else. Also, I could not see an increase in free fatty acids in the hemolymph. Is this because the metabolomics only identifies water-soluble molecules? Since there is an inference made from the metabolomics that the changes in the hemolymph result from the breakdown of material in the fat body, why is there no evidence of increased levels of free fatty acids in the hemolymph? Are they transported in lipoprotein particles or bound to carrier proteins that are not captured in this type of analysis?

Response:

We examined larval muscle, which is known to release amino acids through protein breakdown under

cachectic conditions (Khezri et al., 2021). However, we did not observe classic signs of muscle wasting, such as reduced muscle fiber size under the microscope (Figure S3M–N). It is possible that this assay lacks the sensitivity to detect early or subtle protein degradation occurring within 24 hours of *egr*-expression. Therefore, we cannot rule out that the observed shift in hemolymph amino acid composition may, in part, also be driven by low-level muscle breakdown not detectable by microscopy. Since we observed a more pronounced cachexia-like phenotype in the fat body, we chose to focus our analysis and our conclusions there.

Importantly, we do not report an overall increase in amino acid levels. Rather, we highlight a change in amino acid composition in the hemolymph during the regenerative response, which may reflect both the source of these amino acids and the metabolic demands of regenerating tissues. Notably, the fat body is a major site of larval serum protein expression and storage (proteins that serve as amino acid reservoirs (Lepesant et al., 1982, Valzania et al., 2024)) and their degradation may contribute to the shift observed in the amino acid composition.

Using metabolomics, we measured any detectable metabolite in the hemolymph following tissue damage. As this study focuses on protein translation, we have excluded lipid metabolite data from the representation here. We now clarify the aims in the results and the omission of lipid metabolites in the methods. A separate, ongoing project in our lab currently addresses the role of lipids and lipid metabolism in tissue regeneration and inter-organ signaling, where we will more thoroughly analyse (and interpret changes to) lipid-related metabolites.

4) I was puzzled by their observation that "we observed normal membrane levels of PIP2/3 species in *egr*-expressing discs". If insulin PI3K signaling is reduced in these discs as assessed by increased nuclear FOXO, would you not expect the levels of PIP3 in the membrane to be reduced? Maybe it is difficult to see a decrease with this reagent. Please explain/clarify.

Response:

After thorough discussion with several colleagues, we have decided to remove this dataset from the manuscript. We acknowledge that the results were puzzling and do not align with observations reported by other laboratories. Since this dataset is not central to our main line of argument, we hope the reviewer agrees that its removal is appropriate. Moving forward, we will of course test independent tGPH lines.

5) Can the authors please comment on the effect of some of their manipulations on the size of adult wings? In some ways that provides a good assay for the efficacy of regeneration (more than the quantification of stainings).

Response:

Over the years, we have become cautious in using the analysis of adult wing size as a read-out for the 'success of disc-intrinsic compensatory proliferation'. We believe that other factors can reduce the adult wing size in this set up:

- Altered systemic signals, for example through changes in Dilp8-levels produced by genetically manipulated *egr*-expressing cells, can shorten the regenerative window and mimic reduced proliferation.
- Impaired differentiation of wing disc cells after regeneration can lead to cell death and tissue loss during wing metamorphosis, mimicking reduced proliferation during larval stages.
- Disruption of morphogenetic movements during wing eversion in pupal stages due to 'scaring and epithelial disruption after *egr*-expression' can limit wing expansion, also phenocopying reduced proliferation.
- In heterozygous mutant animals, effects from distant heterozygous tissues such as the fat body, brain, gut, or muscles can affect wing size.

Given these observations, we decided not to include an adult wing size analysis in this manuscript. Additionally, we made the following observations:

To assess the role of Pdk1, we introduced a heterozygous *Pdk1* mutation into the *egr*-expressing background. This resulted in a strongly reduced number of adults emerging upon *egr*-expression – in fact we observed a roughly 4-fold increase in lethality, when compared to the *egr*-only controls, such that often not more than 2 adult flies emerge per experimental vial. The resulting low number of adult wings prevented a statistically meaningful analysis, and the interpretation of this experiment (including the underlying cause of the high lethality) may be limited by any of the factors outlined above.

6) In the discussion, the authors should point out that this model of wounding is associated with an unusually high level of TNF/*eiger* because that is what is used to induce tissue death. Some of these changes might be less relevant in wounds that do not cause dramatic elevation in TNF levels.

Response:

We agree and we have now included a statement about this in the discussion. Accordingly, we recognize that some of the changes we observed under these conditions may not be directly applicable to contexts with lower TNF levels.

At least some of the points that I have brought up suggest that the writing should be a little more careful with appropriate caveats pointed out.

Response:

We hope that at least some of our changes address your concerns.

Minor points (probably typos etc)

Line 141 "Fig 1N" I cannot find Fig. 1N

Response:

Figure 1N is a summary schematic included in the figure. Due to space constraints, we positioned it above Figure 1M.

Line 151 Expression of either" Do you mean expression of *eiger*?

Response:

Thank you! We meant *eiger*.

Figure 2C, D. What is "control". Are these larvae that have had the temperature shift but no *eiger*?

Response:

Apologies for not stating this clearly in the original figure legend. In Figure 2C, 2D and 2E, the control larvae were also subjected to the temperature shift but without *egr* expression, consistent with the other controls used for *egr*-expressing larvae. We have now revised the figure legend to clearly describe both the control and experimental conditions.

Figure 3E. What are the p values?

Response:

Added, Thanks! Now also added to Fig 5 and 6.

Reviewer #3 (Remarks to the Author):

The authors have nicely addressed the issues raised in my original review.

Thank you.

References

- DING, G., XIANG, X., HU, Y., XIAO, G., CHEN, Y., BINARI, R., COMJEAN, A., LI, J., RUSHWORTH, E., FU, Z., MOHR, S. E., PERRIMON, N. & SONG, W. 2021. Coordination of tumor growth and host wasting by tumor-derived Upd3. *Cell Rep*, 36, 109553.
- HARRIS, R. E., STINCHFIELD, M. J., NYSTROM, S. L., MCKAY, D. J. & HARIHARAN, I. K. 2020. Damage-responsive, maturity-silenced enhancers regulate multiple genes that direct regeneration in *Drosophila*. *Elife*, 9.
- HERSPERGER, F., MEYRING, T., WEBER, P., CHHATBAR, C., MONACO, G., DIONNE, M. S., PAESCHKE, K., PRINZ, M., GROSS, O., CLASSEN, A. K. & KIERDORF, K. 2024. DNA damage signaling in *Drosophila* macrophages modulates systemic cytokine levels in response to oxidative stress. *Elife*, 12.
- JIANG, H., PATEL, P. H., KOHLMAIER, A., GRENLEY, M. O., MCEWEN, D. G. & EDGAR, B. A. 2009. Cytokine/Jak/Stat signaling mediates regeneration and homeostasis in the *Drosophila* midgut. *Cell*, 137, 1343-55.
- KHEZRI, R., HOLLAND, P., SCHOBORG, T. A., ABRAMOVICH, I., TAKATS, S., DILLARD, C., JAIN, A., O'FARRELL, F., SCHULTZ, S. W., HAGOPIAN, W. M., QUINTANA, E. M., NG, R., KATHEDER, N. S., RAHMAN, M. M., TELES REIS, J. G., BRECH, A., JASPER, H., RUSAN, N. M., JAHREN, A. H., GOTTLIEB, E. & RUSTEN, T. E. 2021. Host autophagy mediates organ wasting and nutrient mobilization for tumor growth. *EMBO J*, 40, e107336.
- LEPESANT, J. A., LEVINE, M., GAREN, A., LEPESANT-KEJZLARVOA, J., RAT, L. & SOMME-MARTIN, G. 1982. Developmentally regulated gene expression in *Drosophila* larval fat bodies. *J Mol Appl Genet*, 1, 371-83.
- ROMAO, D., MUZZOPAPPA, M., BARRIO, L. & MILAN, M. 2021. The Upd3 cytokine couples inflammation to maturation defects in *Drosophila*. *Curr Biol*, 31, 1780-1787 e6.
- SHIN, M., CHA, N., KORANTENG, F., CHO, B. & SHIM, J. 2020. Subpopulation of Macrophage-Like Plasmacytes Attenuates Systemic Growth via JAK/STAT in the *Drosophila* Fat Body. *Front Immunol*, 11, 63.
- VALZANIA, L., ALAMI, A. & LEOPOLD, P. 2024. A temporal allocation of amino acid resources ensures fitness and body allometry in *Drosophila*. *Dev Cell*, 59, 2277-2286 e6.
- WRIGHT, V. M., VOGT, K. L., SMYTHE, E. & ZEIDLER, M. P. 2011. Differential activities of the *Drosophila* JAK/STAT pathway ligands Upd, Upd2 and Upd3. *Cell Signal*, 23, 920-7.

REVIEWER COMMENTS

Reviewer #1 (Remarks to the Author):

I appreciate the additional data and revisions. However, I remain unconvinced that the core conceptual and mechanistic issues I raised in my previous reviews have been adequately addressed. Below, I highlight the most critical concerns that still require clarification or more rigorous treatment:

1. In my previous comments, I noted that the regions of ablation and knockdown in the Dual Control system were unclear. In response, the authors introduced demarcations that appear arbitrary. A more pressing concern is that Pdk1 or Stat92E inhibition appears to affect OPP signal even outside of the designated knockdown regions. The authors attribute this to a “broader systemic response to damage,” which they claim is described in detail in the manuscript. However, I was unable to find a clear explanation or analysis of this observation. This issue deserves more careful and quantitative evaluation, as it challenges the interpretation of tissue-autonomous effects.

We agree that our original figure annotations did not sufficiently explain how the domains were determined:

- First, we used salm-GAL4, UAS-GFP to approximate the domain in which *eiger* is activated and causes cell death in the Dual Control system (**Reviewer Figure A**). We confirmed that MMP-1 activation coincides with the JNK domain when *eiger* is driven by salm-GAL4 (**Reviewer Figure B**). We concluded that MMP-1 staining and pyknotic nuclei provide two independent markers that allow to track the *egr*-expressing domain in the Dual Control system. Yet, MMP-1 staining is generally graded, diffuse and strongest basally. Therefore, we used a max projection of many sections to create pixel information dense enough to approximate the *egr*-expressing domain (which we apologize for not explaining). Since this is leaving you as a reader with the impression of arbitrariness, we now removed the magenta line and only present DAPI and MMP-1 images, allowing the reader to deduce the *egr*-expressing domain for themselves. However, we describe the required reasoning in figure legends in Figure 5, S5, 6 and S6.
- In the previous manuscript, the cyan outline was derived from experiments, where we had mapped *dve*-GAL4 by UAS-GFP coexpression to stable morphological landmarks that hold - even when *eiger* is expressed (proximal pouch fold on the ventral and dorsal side) (**Reviewer Figure C,D**). To eliminate the impression of arbitrariness, we now remove the cyan line and instead show the UAS-GFP pattern for the control genotype processed in the same tube to distinguish it from RNAi-expressing discs - while ensuring directly comparable fluorescence intensities. The legend explains now how the *dve*-GAL4 domain can be inferred from these data in Figure 5, S5, 6 and S6.
- With respect to your comment about the perceived peripheral OPP downregulation: We describe the peripheral downregulation of protein translation in the manuscript, indeed just not in the context of control vs RNAi disc of the dual control system which we now understand you addressed. We therefore quantified OPP intensities in the (dorsal and ventral) distal hinge in both Pdk1 and Stat92E RNAi discs in single sections. We do not detect a statistically significant reduction in peripheral OPP signal (**Reviewer Figure E,F**; Mean and 95% CI are shown. Statistical significance was tested using a two-tailed Unpaired t-test, except in F' we used a Mann-Whitney Test (control: n=9, *egr*-expressing disc: n=9). We hope that this resolves your concern.

E DCS pouch

E' notum fold (outside of DCS domain)

E'' ventral hinge (outside of DCS domain)

F DCS pouch

F' notum fold (outside of DCS domain)

F'' ventral hinge (outside of DCS domain)

2. A central weakness of the study remains unresolved — the mechanism by which JAK/STAT signaling regulates Pdk1 remains speculative. Despite raising this point in both of my previous reviews, the authors continue to dismiss it as “beyond the scope” of the study. However, given that this pathway forms the conceptual backbone of the manuscript, the absence of even preliminary mechanistic insight (e.g., transcriptional regulation, post-transcriptional control, or intermediary signaling) undermines the strength of the core claims. Without clarification of this regulatory link, the JAK/STAT–PDK1–TORC axis remains correlative and incomplete. At this stage, I am unable to support publication without further clarification.

We apologize if we did appear to be dismissive. In fact, we agree with you that it would be great to understand the upstream mechanism. However, since Pdk-1 is not transcriptionally upregulated, we would have to test a molecular hypothesis based on the regulation of Pdk-1 mRNA translation rates or Pdk-1 protein stability. This would require dedicated molecular tools and biochemical experiments that we cannot provide right now. Our contribution is to provide genetic and functional evidence that Pdk1 upregulation is required for regenerative growth in inflamed tissues and tumors that cause a low insulin environment - an advance that stands independently of the unresolved upstream mechanism.

3. The manuscript proposes that Upd ligands (e.g., Upd3) reach peripheral tissues, yet these same tissues do not exhibit JAK/STAT or Pdk1 activation. This discrepancy is not adequately addressed. The authors suggest that context-dependent responses account for this divergence — a plausible idea, but one that remains speculative without spatially resolved evidence or functional validation. If the proposed model depends on selective activation of Pdk1 by Upd ligands, a clearer mechanistic rationale and supporting data are essential.

To our knowledge, tools enabling the spatial, high-resolution tracking of Unpaired ligands and activation of JAK/STAT effectors in intact wing discs or organ systems are currently unavailable. We therefore cannot do the experiments you are requesting.

4. The authors argue that implicating Pdk1 upregulation in regenerative growth is itself a novel finding. I respectfully disagree. As I have consistently stated, without clarifying the mechanistic link between Upd signaling and Pdk1 regulation, the manuscript does not substantially advance current understanding in the field. In its present form, the findings remain primarily descriptive and offer only incremental novelty in the absence of mechanistic resolution.

We are sorry to hear that but respect your opinion. We truly appreciate your criticism as addressing it over the course of these revisions has really improved our manuscript. However, currently we cannot provide the mechanistic link that you are requesting.